# NUQSGD: Provably Communication-efficient Data-parallel SGD via Nonuniform Quantization

## Abstract

As the size and complexity of models and datasets grow, so does the need for communication-efficient variants of stochastic gradient descent that can be deployed on clusters to perform parallel model training. Alistarh et al. (2017) describe two variants of data-parallel SGD that quantize and encode gradients to lessen communication costs. For the first variant, QSGD, they provide strong theoretical guarantees. For the second variant, which we call QSGDinf, they demonstrate impressive empirical gains for distributed training of large neural networks. Building on their work, we propose an alternative scheme for quantizing gradients and show that it yields stronger theoretical guarantees than exist for QSGD while matching the empirical performance of QSGDinf.

## 1 Introduction

Deep learning is booming thanks to enormous datasets and very large models, leading to the fact that the largest datasets and models can no longer be trained on a single machine. One common solution to this problem is to use distributed systems for training. The most common algorithms underlying deep learning are stochastic gradient descent (SGD) and its variants, which led to a significant amount of research on building and understanding distributed versions of SGD.

Implementations of SGD on distributed systems and data-parallel versions of SGD are scalable and take advantage of multi-GPU systems. Data-parallel SGD, in particular, has received significant attention due to its excellent scalability properties (Zinkevich et al., 2010; Bekkerman et al., 2011; Recht et al., 2011; Dean et al., 2012; Coates et al., 2013; Chilimbi et al., 2014; Li et al., 2014; Duchi et al., 2015; Xing et al., 2015; Zhang et al., 2015; Alistarh et al., 2017). In data-parallel SGD, a large dataset is partitioned among $K$ processors. These processors work together to minimize an objective function. Each processor has access to the current parameter vector of the model. At each SGD iteration, each processor computes an updated stochastic gradient using its own local data. It then shares the gradient update with its peers. The processors collect and aggregate stochastic gradients to compute the updated parameter vector.

Increasing the number of processing machines reduces the computational costs significantly. However, the communication costs to share and synchronize huge gradient vectors and parameters increases dramatically as the size of the distributed systems grows. Communication costs may thwart the anticipated benefits of reducing computational costs. Indeed, in practical scenarios, the communication time required to share stochastic gradients and parameters is the main performance bottleneck (Recht et al., 2011; Li et al., 2014; Seide et al., 2014; Strom, 2015; Alistarh et al., 2017). Reducing communication costs in data-parallel SGD is an important problem.

One promising solution to the problem of reducing communication costs of data-parallel SGD is gradient compression, *e.g.,* through gradient quantization (Dean et al., 2012; Seide et al., 2014; Sa et al., 2015; Gupta et al., 2015; Abadi et al., 2016; Zhou et al., 2016; Alistarh et al., 2017; Wen et al., 2017; Bernstein et al., 2018). (This should not be confused with weight quantization/sparsification, as studied by Wen et al. (2016); Hubara et al. (2016); Park et al. (2017); Wen et al. (2017), which we do not discuss here.) Unlike full-precision data-parallel SGD, where each processor is required to broadcast its local gradient in full-precision, *i.e.,* transmit and receive huge full-precision vectors at each iteration, quantization requires each processor to transmit only a few communication bits per iteration for each component of the stochastic gradient.

One popular such proposal for communication-compression is quantized SGD (QSGD), due to Alistarh et al. (2017). In QSGD, stochastic gradient vectors are normalized to have unit $L^2$ norm,

and then compressed by quantizing each element to a uniform grid of quantization levels using a randomized method. While most lossy compression schemes do not provide convergence guarantees, QSGD's quantization scheme, is designed to be unbiased, which implies that the quantized stochastic gradient is itself a stochastic gradient, only with higher variance determined by the dimension and number of quantization levels. As a result, Alistarh et al. (2017) are able to establish a number of theoretical guarantees for QSGD, including that it converges under standard assumptions. By changing the number of quantization levels, QSGD allows the user to trade-off communication bandwidth and convergence time.

Despite their theoretical guarantees based on quantizing after $L^2$ normalization, Alistarh et al. opt to present empirical results using $L^\infty$ normalization. We call this variation QSGDinf. While the empirical performance of QSGDinf is strong, their theoretical guarantees on the number of bits transmitted no longer apply. Indeed, in our own empirical evaluation of QSGD, we find the variance induced by quantization is substantial, and the performance is far from that of SGD and QSGDinf.

Given the popularity of this scheme, it is natural to ask one can obtain guarantees as strong as those of QSGD while matching the practical performance of the QSGDinf heuristic. In this work, we answer this question in the affirmative by providing a new quantization scheme which fits into QSGD in a way that allows us to establish stronger theoretical guarantees on the variance, bandwidth, and cost to achieve a prescribed gap. Instead of QSGD's uniform quantization scheme, we use an unbiased nonuniform logarithmic scheme, similar to those introduced in telephony systems for audio compression (Cattermole, 1969). We call the resulting algorithm *nonuniformly quantized stochastic gradient descent* (NUQSGD). Like QSGD, NUQSGD is a quantized data-parallel SGD algorithm with strong theoretical guarantees that allows the user to trade off communication costs with convergence speed. Unlike QSGD, NUQSGD has strong empirical performance on deep models and large datasets, matching that of QSGDinf. In particular, we provide a new efficient implementation for these schemes using a modern computational framework (Pytorch), and benchmark it on classic large-scale image classification tasks.

The intuition behind the nonuniform quantization scheme underlying NUQSGD is that, after $L^2$ normalization, many elements of the normalized stochastic gradient will be near-zero. By concentrating quantization levels near zero, we are able to establish stronger bounds on the excess variance. In the overparametrized regime of interest, these bounds decrease rapidly as the number of quantization levels increases. Combined with a bound on the expected code-length, we obtain a bound on the total communication costs of achieving an expected suboptimality gap. The resulting bound is slightly stronger than the one provided by QSGD.

To study how quantization affects convergence on state-of-the-art deep models, we compare NUQSGD, QSGD, and QSGDinf, focusing on training loss, variance, and test accuracy on standard deep models and large datasets. Using the same number of bits per iteration, experimental results show that NUQSGD has smaller variance than QSGD, as expected by our theoretical results. This smaller variance also translates to improved optimization performance, in terms of both training loss and test accuracy. We also observe that NUQSGD matches the performance of QSGDinf in terms of variance and loss/accuracy. Further, our distributed implementation shows that the resulting algorithm considerably reduces communication cost of distributed training, without adversely impacting accuracy. Our empirical results show that NUQSGD can provide faster end-to-end parallel training relative to data-parallel SGD, QSGD, and Error-Feedback SignSGD (Karimireddy et al., 2019) on the ImageNet dataset.

**Summary of Contributions.**

- We establish stronger theoretical guarantees for the excess variance and communication costs of our gradient quantization method than those available for QSGD's uniform quantization method.

- We then establish stronger convergence guarantees for the resulting algorithm, NUQSGD, under standard assumptions.

- We demonstrate that NUQSGD has strong empirical performance on deep models and large datasets, both in terms of accuracy and scalability. Thus, NUQSGD closes the gap between the theoretical guarantees of QSGD and the empirical performance of QSGDinf.

## 1.1 RELATED WORK

Seide et al. (2014) proposed signSGD, an efficient heuristic scheme to reduce communication costs drastically by quantizing each gradient component to two values. Bernstein et al. (2018) later provided convergence guarantees for signSGD. Note that the quantization employed by signSGD is not unbiased, and so a new analysis was required. As the number of levels is fixed, SignSGD does not provide any trade-off between communication costs and convergence speed.

Sa et al. (2015) introduced Buckwild!, a lossy compressed SGD with convergence guarantees. The authors provided bounds on the error probability of SGD, assuming convexity and gradient sparsity.

Wen et al. (2017) proposed TernGrad, a stochastic quantization scheme with three levels. TernGrad also significantly reduces communication costs and obtains reasonable accuracy with a small degradation to performance compared to full-precision SGD. Convergence guarantees for TernGrad rely on a nonstandard gradient norm assumption. As discussed, Alistarh et al. (2017) proposed QSGD, a more general stochastic quantization scheme, for which they provide both theoretical guarantees and experimental validation (although for different variants of the same algorithm). We note that their implementation was only provided in Microsoft CNTK; by contrast, here we provide a more generic implementation in Horovod (Sergeev and Del Balso, 2018), a communication back-end which can support a range of modern frameworks such as Tensorflow, Keras, Pytorch, and MXNet.

NUQSGD uses a logarithmic quantization scheme. Such schemes have long been used in telephony systems for audio compression (Cattermole, 1969). Logarithmic quantization schemes have appeared in other contexts recently: Hou and Kwok (2018) studied weight distributions of long short-term memory networks and proposed to use logarithm quantization for network compression. Zhang et al. (2017) proposed a gradient compression scheme and introduced an optimal quantization scheme, but for the setting where the points to be quantized are known in advance. As a result, their scheme is not applicable to the communication setting of quantized data-parallel SGD.

## 2 PRELIMINARIES: DATA-PARALLEL SGD AND CONVERGENCE

We consider a high-dimensional machine learning model, parametrized by a vector $\mathbf{w} \in \mathbb{R}^d$. Let $\Omega \subseteq \mathbb{R}^d$ denote a closed and convex set. Our objective is to minimize $f : \Omega \to \mathbb{R}$, which is an unknown, differentiable, convex, and $\beta$-smooth function. The following summary is based on (Alistarh et al., 2017).

Recall that a function $f$ is $\beta$-**smooth** if, for all $\mathbf{u}, \mathbf{v} \in \Omega$, we have $\|\nabla f(\mathbf{u}) - \nabla f(\mathbf{v})\| \leq \beta \|\mathbf{u} - \mathbf{v}\|$, where $\|\cdot\|$ denotes the Euclidean norm. Let $(\mathscr{S}, \Sigma, \mu)$ be a probability space (and let $\mathbb{E}$ denote expectation). Assume we have access to stochastic gradients of $f$, *i.e.,* we have access to a function $g : \Omega \times \mathscr{S} \to \mathbb{R}^d$ such that, if $s \sim \mu$, then $\mathbb{E}[g(\mathbf{w}, s)] = \nabla f(\mathbf{w})$ for all $\mathbf{w} \in \Omega$. In the rest of the paper, we let $g(\mathbf{w})$ denote the stochastic gradient for notational simplicity. The update rule for conventional full-precision projected SGD is $\mathbf{w}_{t+1} = \mathbf{P}_\Omega(\mathbf{w}_t - \alpha g(\mathbf{w}_t))$, where $\mathbf{w}_t$ is the current parameter input, $\alpha$ is the learning rate, and $\mathbf{P}_\Omega$ is the Euclidean projection onto $\Omega$.

We say the stochastic gradient has a **second-moment upper bound** $B$ when $\mathbb{E}[\|g(\mathbf{w})\|^2] \leq B$ for all $\mathbf{w} \in \Omega$. Similarly, the stochastic gradient has a **variance upper bound** $\sigma^2$ when $\mathbb{E}[\|g(\mathbf{w}) - \nabla f(\mathbf{w})\|^2] \leq \sigma^2$ for all $\mathbf{w} \in \Omega$. Note that a second-moment upper bound implies a variance upper bound, because the stochastic gradient is unbiased.

We have classical convergence guarantees for conventional full-precision SGD given access to stochastic gradients at each iteration:

**Theorem 1** (Bubeck 2015, Theorem 6.3). *Let $f : \Omega \to \mathbb{R}$ denote a convex and $\beta$-smooth function and let $R^2 \triangleq \sup_{\mathbf{w} \in \Omega} \|\mathbf{w} - \mathbf{w}_0\|^2$. Suppose that the projected SGD update is executed for $T$ iterations with $\alpha = 1/(\beta + 1/\gamma)$ where $\gamma = r\sqrt{2/T}/\sigma$. Given repeated and independent access to stochastic gradients with a variance upper bound $\sigma^2$, projected SGD satisfies*

$$\mathbb{E}\left[f\left(\frac{1}{T}\sum_{t=0}^{T}\mathbf{w}_t\right)\right] - \min_{\mathbf{w}\in\Omega} f(\mathbf{w}) \leq R\sqrt{\frac{2\sigma^2}{T}} + \frac{\beta R^2}{T}. \tag{1}$$

Minibatched (with larger batch sizes) and data-parallel SGD are two common SGD variants used in practice to reduce variance and improve computational efficiency of conventional SGD.

**Input:** local data, local copy of the parameter vector $\mathbf{w}_t$, learning rate $\alpha$, and $K$

**1 for** $t = 1$ **to** $T$ **do**

**2**     **for** $i = 1$ **to** $K$ **do** // each transmitter processor (in parallel)

**3**        Compute $g_i(\mathbf{w}_t)$ ; // stochastic gradient

**4**        Encode $c_{i,t} \leftarrow \text{ENCODE}\big(g_i(\mathbf{w}_t)\big)$;

**5**        Broadcast $c_{i,t}$ to all processors;

**6**     **for** $l = 1$ **to** $K$ **do** // each receiver processor (in parallel)

**7**        **for** $i = 1$ **to** $K$ **do** // each transmitter processor

**8**           Receive $c_{i,t}$ from processor $i$ for each $i$;

**9**           Decode $\hat{g}_i(\mathbf{w}_t) \leftarrow \text{DECODE}\big(c_{i,t}\big)$;

**10**        Aggregate $\mathbf{w}_{t+1} \leftarrow \mathbf{P}_\Omega(\mathbf{w}_t - \frac{\alpha}{K}\sum_{i=1}^K \hat{g}_i(\mathbf{w}_t))$;

**Algorithm 1:** Data-parallel (synchronized) SGD.

Following (Alistarh et al., 2017), we consider data-parallel SGD, a synchronous distributed framework consisting of $K$ processors that partition a large dataset among themselves. This framework models real-world systems with multiple GPU resources. Each processor keeps a local copy of the parameter vector and has access to independent and private stochastic gradients of $f$.

At each iteration, each processor computes its own stochastic gradient based on its local data and then broadcasts it to all peers. Each processor receives and aggregates the stochastic gradients from all peers to obtain the updated parameter vector. In detail, the update rule for full-precision data-parallel SGD is $\mathbf{w}_{t+1} = \mathbf{P}_\Omega(\mathbf{w}_t - \frac{\alpha}{K}\sum_{l=1}^K \overline{g}_l(\mathbf{w}_t))$ where $\overline{g}_l(\mathbf{w}_t)$ is the stochastic gradient computed and broadcasted by processor $l$. Provided that $\overline{g}_l(\mathbf{w}_t)$ is a stochastic gradient with a variance upper bound $\sigma^2$ for all $l$, then $\frac{1}{K}\sum_{l=1}^K \overline{g}_l(\mathbf{w}_t)$ is a stochastic gradient with a variance upper bound $\frac{\sigma^2}{K}$. Thus, aggregation improves convergence of SGD by reducing the first term of the upper bound in (1). Assume each processor computes a minibatch gradient of size $B$. Then, this update rule is essentially a minibatched update with size $BK$.

Data-parallel SGD is described in Algorithm 1. Full-precision data-parallel SGD is a special case of Algorithm 1 with identity encoding and decoding mappings. Otherwise, the decoded stochastic gradient $\hat{g}_i(\mathbf{w}_t)$ is likely to be different from the original local stochastic gradient $g_i(\mathbf{w}_t)$.

By Theorem 1, we have the following convergence guarantees for full-precision data-parallel SGD:

**Corollary 1** (Alistarh et al. 2017, Corollary 2.2). *Let $f$, $R$, and $\gamma$ be as defined in Theorem 1 and let $\varepsilon > 0$. Suppose that the projected SGD update is executed for $T$ iterations with $\alpha = 1/(\beta + \sqrt{K}/\gamma)$ on $K$ processors, each with access to independent stochastic gradients of $f$ with a second-moment bound $B$. The smallest $T$ for the full-precision data-parallel SGD that guarantees $\mathbb{E}\big[f(\frac{1}{T}\sum_{t=0}^T \mathbf{w}_t)\big] - \min_{\mathbf{w}\in\Omega} f(\mathbf{w}) \leq \varepsilon$ is $T_\varepsilon = O\big(R^2 \max(\frac{2B}{K\varepsilon^2}, \frac{\beta}{\varepsilon})\big)$.*

## 3 NONUNIFORMLY QUANTIZED STOCHASTIC GRADIENT DESCENT

Data-parallel SGD reduces computational costs significantly. However, the communication costs of broadcasting stochastic gradients is the main performance bottleneck in large-scale distributed systems. In order to reduce communication costs and accelerate training, Alistarh et al. (2017) introduced a compression scheme that produces a compressed and unbiased stochastic gradient, suitable for use in SGD.

At each iteration of QSGD, each processor broadcasts an encoding of its own compressed stochastic gradient, decodes the stochastic gradients received from other processors, and sums all the quantized vectors to produce a stochastic gradient. In order to compress the gradients, every coordinate (with respect to the standard basis) of the stochastic gradient is normalized by the Euclidean norm of the gradient and then stochastically quantized to one of a small number quantization levels distributed uniformly in the unit interval. The stochasticity of the quantization is necessary to not introduce bias.

Alistarh et al. (2017) give a simple argument that provides a *lower* bound on the number of coordinates that are quantized to zero in expectation. Encoding these zeros efficiently provides communication savings at each iteration. However, the cost of their scheme is greatly increased variance in the gradient, and thus slower overall convergence. In order to optimize overall performance, we must balance communication savings with variance.

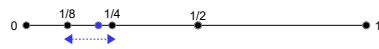

Figure 1: An example of nonuniform stochastic quantization with $s = 3$. The point between the arrows represents the value of the normalized coordinate.

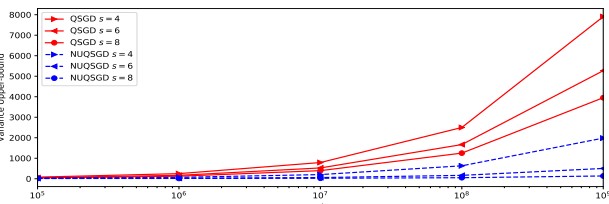

Figure 2: Variance upper bounds.

By simple counting arguments, the distribution of the (normalized) coordinates cannot be uniform. Indeed, this is the basis of the lower bound on the number of zeros. These arguments make no assumptions on the data distribution, and rely entirely on the fact that the quantities being quantized are the coordinates of a unit-norm vector. Uniform quantization does not capture the properties of such vectors, leading to substantial gradient variance.

### 3.1 NONUNIFORM QUANTIZATION

In this paper, we propose and study a new scheme to quantize normalized gradient vectors. Instead of uniformly distributed quantization levels, as proposed by Alistarh et al. (2017), we consider quantization levels that are nonuniformly distributed in the unit interval, as depicted in Figure 1. In order to obtain a quantized gradient that is suitable for SGD, we need the quantized gradient to remain unbiased. Alistarh et al. (2017) achieve this via a randomized quantization scheme, which can be easily generalized to the case of nonuniform quantization levels.

Using a carefully parametrized generalization of the unbiased quantization scheme introduced by Alistarh et al., we can control both the cost of communication and the variance of the gradient. Compared to a uniform quantization scheme, our scheme reduces quantization error and variance by better matching the properties of normalized vectors. In particular, by increasing the number of quantization levels near zero, we obtain a stronger variance bound. Empirically, our scheme also better matches the distribution of normalized coordinates observed on real datasets and networks.

We now describe the nonuniform quantization scheme: Let $s \in \{1, 2, \cdots\}$ be the number of internal quantization levels, and let $\mathscr{L} = (l_0, l_1, \cdots, l_{s+1})$ denote the sequence of quantization levels, where $l_0 = 0 < l_1 < \cdots < l_{s+1} = 1$. For $r \in [0, 1]$, let $\tilde{s}(r)$ and $p(r)$ satisfy $l_{\tilde{s}(r)} \leq r \leq l_{\tilde{s}(r)+1}$ and $r = \left(1 - p(r)\right) l_{\tilde{s}(r)} + p(r) l_{\tilde{s}(r)+1}$, respectively. Define $\tau(r) = l_{\tilde{s}(r)+1} - l_{\tilde{s}(r)}$. Note that $\tilde{s}(r) \in \{0, 1, \cdots, s\}$.

**Definition 1.** *The nonuniform quantization of a vector $\mathbf{v} \in \mathbb{R}^d$ is*

$$Q_s(\mathbf{v}) \triangleq [Q_s(v_1), \cdots, Q_s(v_d)]^T \quad where \quad Q_s(v_i) = \|\mathbf{v}\| \cdot \text{sign}(v_i) \cdot h_i(\mathbf{v}, s) \tag{2}$$

*where, letting $r_i = |v_i| / \|\mathbf{v}\|$, the $h_i(\mathbf{v}, s)$'s are independent random variables such that $h_i(\mathbf{v}, s) = l_{\tilde{s}(r_i)}$ with probability $1 - p(r_i)$ and $h_i(\mathbf{v}, s) = l_{\tilde{s}(r_i)+1}$ otherwise.*

We note that the distribution of $h_i(\mathbf{v}, s)$ satisfies $\mathbb{E}[h_i(\mathbf{v}, s)] = r_i$ and achieves the minimum variance over all distributions that satisfy $\mathbb{E}[h_i(\mathbf{v}, s)] = r_i$ with support $\mathscr{L}$. In the following, we focus on a special case of nonuniform quantization with $\hat{\mathscr{L}} = (0, 1/2^s, \cdots, 2^{s-1}/2^s, 1)$ as the quantization levels.

The intuition behind this quantization scheme is that it is very unlikely to observe large values of $r_i$ in the stochastic gradient vectors of machine learning models. Stochastic gradients are observed to be dense vectors (Bernstein et al., 2018). Hence, it is natural to use fine intervals for small $r_i$ values to reduce quantization error and control the variance.

After quantizing the stochastic gradient with a small number of discrete levels, each processor must encode its local gradient into a binary string for broadcasting. We describe this encoding in Appendix A.

## 4 THEORETICAL GUARANTEES

In this section, we provide theoretical guarantees for NUQSGD, giving variance and code-length bounds, and using these in turn to compare NUQSGD and QSGD. Please note that the proofs of Theorems 2, 3, 4, and 5 are provided in Appendices B, C, D, and E respectively.

**Theorem 2** (Variance bound). *Let $\mathbf{v} \in \mathbb{R}^d$. The nonuniform quantization of $\mathbf{v}$ satisfies $\mathbb{E}[Q_s(\mathbf{v})] = \mathbf{v}$. Furthermore, provided that $s \leq \log(d)/2$, we have*

$$\mathbb{E}[\|Q_s(\mathbf{v}) - \mathbf{v}\|^2] \leq \varepsilon_Q \|\mathbf{v}\|^2 \tag{3}$$

*where $\varepsilon_Q = \min\{\min\{2^{-2s}(d - 2^{2s}), 2^{-s}\sqrt{d - 2^{2s}}\} + O(s), d/3(2^{-2s+1} + 1)\}$.*

The result in Theorem 2 implies that if $g(\mathbf{w})$ is a stochastic gradient with a second-moment bound $\eta$, then $Q_s(g(\mathbf{w}))$ is a stochastic gradient with a variance upper bound $\varepsilon_Q \eta$. In the range of interest where $d$ is sufficiently large, *i.e.*, $s = o(\log(d))$, the variance upper bound decreases with the number of quantization levels. To obtain this data-independent bound, we establish upper bounds on the number of coordinates of $\mathbf{v}$ falling into intervals defined by $\hat{\mathcal{L}}$.

**Theorem 3** (Code-length bound). *Let $\mathbf{v} \in \mathbb{R}^d$. Provided $d$ is large enough to ensure $2^{2s} + \sqrt{d}2^s \leq d/e$, the expectation $\mathbb{E}[|\text{ENCODE}(\mathbf{v})|]$ of the number of communication bits needed to transmit $Q_s(\mathbf{v})$ is bounded above by*

$$N_Q = C + 3n_{s,d} + (1 + o(1))n_{s,d}\log\left(\frac{d}{n_{s,d}}\right) + (1 + o(1))n_{s,d}\log\log\left(\frac{8(2^{2s} + d)}{n_{s,d}}\right) \tag{4}$$

*where $C = b - (1 + o(1))$ and $n_{s,d} = 2^{2s} + 2^s\sqrt{d}$.*

Theorem 3 provides a bound on the expected number of communication bits to encode the quantized stochastic gradient. Note that $2^{2s} + \sqrt{d}2^s \leq d/e$ is a mild assumption in practice. As one would expect, the bound, (4), increases monotonically in $d$ and $s$. In the sparse case, if we choose $s = o(\log d)$ levels, then the upper bound on the expected code-length is $O\left(2^s\sqrt{d}\log\left(\frac{\sqrt{d}}{2^s}\right)\right)$.

Combining the upper bounds above on the variance and code-length, Corollary 1 implies the following guarantees for NUQSGD:

**Theorem 4** (NUQSGD for smooth convex optimization). *Let $f$ and $R$ be defined as in Theorem 1, let $\varepsilon_Q$ be defined as in Theorem 2, let $\varepsilon > 0$, $\hat{B} = (1 + \varepsilon_Q)B$, and let $\gamma > 0$ be given by $\gamma^2 = 2R^2/(\hat{B}T)$. With ENCODE and DECODE defined as in Appendix A, suppose that Algorithm 1 is executed for $T$ iterations with a learning rate $\alpha = 1/(\beta + \sqrt{K}/\gamma)$ on $K$ processors, each with access to independent stochastic gradients of $f$ with a second-moment bound $B$. Then $T_\varepsilon = O\left(\max\left(\frac{2\hat{B}}{K\varepsilon^2}, \frac{\beta}{\varepsilon}\right)R^2\right)$ iterations suffice to guarantee $\mathbb{E}\left[f\left(\frac{1}{T}\sum_{t=0}^T \mathbf{w}_t\right)\right] - \min_{\mathbf{w}\in\Omega} f(\mathbf{w}) \leq \varepsilon$. In addition, NUQSGD requires at most $N_Q$ communication bits per iteration in expectation.*

On nonconvex problems, (weaker) convergence guarantees can be established along the lines of, e.g., (Ghadimi and Lan, 2013, Theorem 2.1).

**NUQSGD vs QSGD.** How do QSGD and NUQSGD compare in terms of bounds on the expected number of communication bits required to achieve a given suboptimality gap $\varepsilon$? The quantity that controls our guarantee on the convergence speed in both algorithms is the variance upper bound, which in turn is controlled by the quantization schemes. Note that the number of quantization levels, $s$, is usually a small number in practice. On the other hand, the dimension, $d$, can be very large, especially in overparameterized networks. In Figure 2, we show that the quantization scheme underlying NUQSGD results in substantially smaller variance upper bounds for plausible ranges of $s$ and $d$. Note that these bounds do not make any assumptions on the dataset or the structure of the network.

For any (nonrandom) number of iterations $T$, an upper bound, $\overline{N}_A$, holding uniformly over iterations $k \leq T$ on the expected number of bits used by an algorithm $A$ to communicate the gradient on iteration $k$, yields an upper bound $T\overline{N}_A$, on the expected number of bits communicated over $T$ iterations by algorithm $A$. Taking $T = T_{A,\varepsilon}$ to be the (minimum) number of iterations needed to guarantee an expected suboptimality gap of $\varepsilon$ based on the properties of $A$, we obtain an upper bound, $\zeta_{A,\varepsilon} = T_{A,\varepsilon}\overline{N}_A$, on the expected number of bits of communicated on a run expected to achieve a suboptimality gap of at most $\varepsilon$.

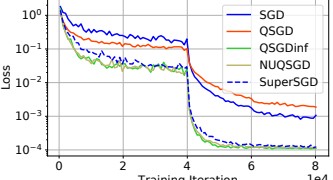 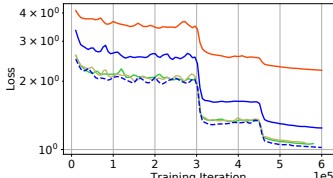 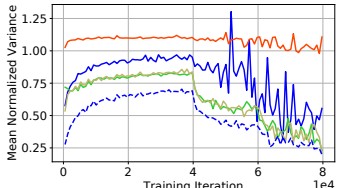

Figure 3: Training loss on CIFAR10 (left) and ImageNet (middle) for ResNet models. QSGD, QSGDinf, and NUQSGD are trained by simulating the quantization and dequantizing of the gradients from 8-GPUs. On CIFAR10, SGD refers to the single-GPU training versus on Imagenet it refers to 2-GPU setup in the original ResNet paper. SGD is shown to highlight the significance of the gap between QSGD and QSGDinf. SuperSGD refers to simulating full-precision distributed training without quantization. SuperSGD is impractical in scenarios with limited bandwidth. (Right) Estimated normalized variance on CIFAR10 on the trajectory of single-GPU SGD. Variance is measured for fixed model snapshots during training. Notice that the variance for NUQSGD and QSGDinf is lower than SGD for almost all the training and it decreases after the learning rate drops.

**Theorem 5** (Expected number of communication bits). *Provided that $s = o(\log(d))$ and $\frac{2\hat{B}}{K\varepsilon^2} > \frac{\beta}{\varepsilon}$, $\zeta_{\text{NUQSGD},\varepsilon} = O\big(\frac{1}{\varepsilon^2}\sqrt{d(d-2^{2s})}\log\big(\frac{\sqrt{d}}{2^s}\big)\big)$ and $\zeta_{\text{QSGD},\varepsilon} = O(\frac{1}{\varepsilon^2}d\log\sqrt{d})$.*

Focusing on the dominant terms in the expressions of overall number of communication bits required to guarantee a suboptimality gap of $\varepsilon$, we observe that NUQSGD provides slightly stronger guarantees. Note that our stronger guarantees come without any assumption about the data.

## 5 EXPERIMENTAL EVALUATION

In this section, we examine the practical performance of NUQSGD in terms of both convergence (accuracy) and speedup. The goal is to empirically show that NUQSGD can provide the same performance and accuracy compared to the QSGDInf heuristic, which has no theoretical compression guarantees. For this, we implement and test these three methods (NUQSGD, QSGD, and QSGDInf), together with the distributed full-precision SGD baseline, which we call SuperSGD. We split our study across two axes: first, we examine the convergence of the methods and their induced variance. Second, we provide an efficient implementation of all four methods in Pytorch using the Horovod communication back-end (Sergeev and Del Balso, 2018), adapted to efficiently support quantization, and examine speedup relative to the full-precision baseline. We investigate the impact of quantization on training performance by measuring loss, variance, accuracy, and speedup for ResNet models (He et al., 2016) applied to ImageNet (Deng et al., 2009) and CIFAR10 (Krizhevsky).

We evaluate these methods on two image classification datasets: ImageNet and CIFAR10. We train ResNet110 on CIFAR10 and ResNet18 on ImageNet with mini-batch size 128 and base learning rate 0.1. In all experiments, momentum and weight decay are set to 0.9 and $10^{-4}$, respectively. The bucket size and the number of quantization bits are set to 8192 and 4, respectively. We observe similar results in experiments with various bucket sizes and number of bits. We simulate a scenario with $k$ GPUs for all three quantization methods by estimating the gradient from $k$ independent mini-batches and aggregating them after quantization and dequantization.

In Figure 3 (left and middle), we show the training loss with 8 GPUs. We observe that NUQSGD and QSGDinf improve training loss compared to QSGD on ImageNet. We observe significant gap in training loss on CIFAR10 where the gap grows as training proceeds. We also observe similar performance gaps in test accuracy (provided in Appendix F). In particular, unlike NUQSGD, QSGD does not achieve test accuracy of full-precision SGD. Figure 3 (right) shows the mean normalized variance of the gradient (defined in Appendix F) versus training iteration on the trajectory of single-GPU SGD on CIFAR10. These observations validate our theoretical results that NUQSGD has smaller variance for large models with small number of quantization bits.

**Efficient Implementation and Speedup.** To examine speedup behavior, we implemented all quantization methods in Horovod (Sergeev and Del Balso, 2018), a communication back-end supporting Pytorch, Tensorflow and MXNet. Doing so efficiently requires non-trivial refactoring of this framework, since it does not support communication compression—our framework will be open-sourced

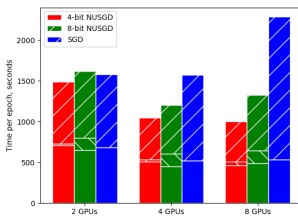 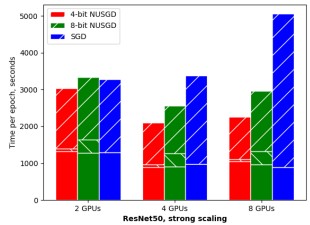 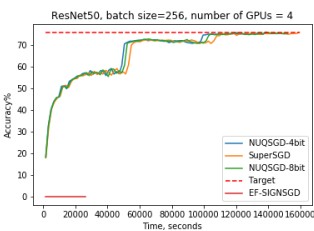

Figure 4: Scalability behavior for NUQSGD versus the full-precision baseline when training ResNet34 and ResNet50 on ImageNet. The ResNet34 graph examines *strong scaling* (left), splitting a global batch of size 256 onto the available GPUs, whereas the ResNet50 graph examines strong scaling (middle) keeping a fixed per-GPU batch size of 16. Each time bar is split into computation (bottom), encoding cost (middle), and transmission cost (top). Notice the significant negative scalability of the SGD baseline in both scenarios. By contrast, the 4-bit communication-compressed implementation achieves positive scaling, while the 8-bit variant stops scaling between 4 and 8 nodes due to the higher communication and encoding costs. End-to-end training time for ResNet50/ImageNet for NUQSGD and EF-SignSGD versus the SuperSGD baseline (right).

upon publication. Our implementation diverges slightly from the theoretical analysis. First, Horovod applies "tensor fusion" to multiple layers, by merging the resulting gradient tensors for more efficient transmission. This causes the gradients for different layers to be quantized together, which can lead to loss of accuracy (due to e.g. different normalization factors across the layers). We addressed this by tuning the way in which tensor fusion is applied to the layers such that it minimizes the accuracy loss. Second, we noticed that quantizing the gradients corresponding to the biases has a significant adverse effect on accuracy; since the communication impact of biases is negligible, we transmit them at full precision. We apply this for all methods considered. Finally, for efficiency reasons, we directly pack the quantized values into 32-bit numbers, without additional encoding. We implemented compression and de-compression via efficient CUDA kernels.

Our baselines are full-precision SGD (SuperSGD), Error-Feedback SignSGD (Karimireddy et al., 2019), and the QSGDinf heuristic, which we compare against the 4-bit and 8-bit NUQSGD variants executing the same pattern. The implementation of the QSGDinf heuristic provides almost identical convergence numbers, and is sometimes omitted for visibility. (QSGD yields inferior convergence on this dataset and is therefore omitted.) All variants are implemented using a standard all-to-all reduction pattern. Figures 4 (left), (middle) show the execution time per epoch for ResNet34 and ResNet50 models on ImageNet, on a cluster machine with 8 NVIDIA 2080 Ti GPUs, for the hyper-parameter values quoted above. The results confirm the efficiency and scalability of the compressed variant, mainly due to the reduced communication volume. We note that the overhead of compression and decompression is less than 1% of the batch computation time for NUQSGD.

Figure 4 (right) presents end-to-end speedup numbers (time versus accuracy) for ResNet50/ImageNet, executed on 4 GPUs, under the same hyperparameter settings as the full-precision baseline, with bucket size 512. First, notice that NUQSGD variants match the target accuracy of the 32-bit model, with non-trivial speedup over the standard data-parallel variant, directly proportional to the per-epoch speedup. The QSGDinf heuristic yields similar accuracy and performance, and is therefore omitted. Second, we found that unfortunately EF-SignSGD does not converge under these standard hyperparameter settings. To address this issue, we performed a non-trivial amount of hyperparameter tuning for this algorithm: in particular, we found that the scaling factors and the bucket size must be carefully adjusted for convergence on ImageNet. We were able to recover full accuracy with EF-SignSGD on ResNet50, but that the cost of quantizing into buckets of size 64. Unfortunately, in this setting the algorithm transmits a non-trivial amount of scaling data, and the GPU implementation becomes less efficient due to error computation and reduced parallelism. The end-to-end speedup of this tuned variant is inferior to NUQSGD-4bit, and only slightly superior to that of NUQSGD-8bit. Please see Figure 9 in the Appendix and the accompanying text for details.

## 6  CONCLUSIONS

We study data-parallel and communication-efficient version of stochastic gradient descent. Building on QSGD (Alistarh et al., 2017), we study a nonuniform quantization scheme. We establish

upper bounds on the variance of nonuniform quantization and the expected code-length. In the overparametrized regime of interest, the former decreases as the number of quantization levels increases, while the latter increases with the number of quantization levels. Thus, this scheme provides a trade-off between the communication efficiency and the convergence speed. We compare NUQSGD and QSGD in terms of their variance bounds and the expected number of communication bits required to meet a certain convergence error, and show that NUQSGD provides stronger guarantees. Experimental results are consistent with our theoretical results and confirm that NUQSGD matches the performance of QSGDinf when applied to practical deep models and datasets including ImageNet. Thus, NUQSGD closes the gap between the theoretical guarantees of QSGD and empirical performance of QSGDinf. One limitation of our study which we aim to address in future work is that we focus on all-to-all reduction patterns, which interact easily with communication compression. In particular, we aim to examine the interaction between more complex reduction patterns, such as ring-based reductions (Hannun et al., 2014), which may yield superior performance in bandwidth-bottlenecked settings, but which interact with communication-compression in non-trivial ways, since they may lead a gradient to be quantized at each reduction step.

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

## A  ENCODING

**Encoding:**
1 Place a 0 at the end of the string;
2 **if** $N == 1$ **then**
3 $\quad$ Stop;
4 **else**
5 $\quad$ Prepend binary$(N)$ to the beginning;
6 $\quad$ Let $N'$ denote # bits prepended minus 1;
7 $\quad$ Encode $N'$ recursively;
**Decoding:**
8 Start with $N = 1$;
9 **if** *the next bit* $== 0$ **then**
10 $\quad$ Stop and return $N$;
11 **else**
12 $\quad$ Read that bit plus $N$ following bits;
13 $\quad$ Update $N$;
**Algorithm 2:** Elias recursive coding produces a bit string encoding of positive integers.

By inspection, the quantized gradient $Q_s(\mathbf{v})$ is determined by the tuple $(\|\mathbf{v}\|, \boldsymbol{\rho}, \mathbf{h})$, where $\|\mathbf{v}\|$ is the norm of the gradient, $\boldsymbol{\rho} \triangleq [\text{sign}(v_1), \cdots, \text{sign}(v_d)]^T$ is the vector of signs of the coordinates $v_i$'s, and $\mathbf{h} \triangleq [h_1(\mathbf{v}, s), \cdots, h_d(\mathbf{v}, s)]^T$ are the quantizations of the normalized coordinates. We can describe the ENCODE function (for Algorithm 1) in terms of the tuple $(\|\mathbf{v}\|, \boldsymbol{\rho}, \mathbf{h})$ and an encoding/decoding scheme $\text{ERC} : \{1, 2, \cdots\} \to \{0, 1\}^*$ and $\text{ERC}^{-1} : \{0, 1\}^* \to \{1, 2, \cdots\}$ for encoding/decoding positive integers.

The encoding, ENCODE$(\mathbf{v})$, of a stochastic gradient is as follows: We first encode the norm $\|\mathbf{v}\|$ using $b$ bits where, in practice, we use standard 32-bit floating point encoding. We then proceed in rounds, $r = 0, 1, \cdots$. On round $r$, having transmitted all nonzero coordinates up to and including $t_r$, we transmit $\text{ERC}(i_r)$ where $t_{r+1} = t_r + i_r$ is either (i) the index of the first nonzero coordinate of $\mathbf{h}$ after $t_r$ (with $t_0 = 0$) or (ii) the index of the last nonzero coordinate. In the former case, we then transmit one bit encoding the sign $\rho_{t_{r+1}}$, transmit $\text{ERC}(\log(2^{s+1}h_{t_{r+1}}))$, and proceed to the next round. In the latter case, the encoding is complete after transmitting $\rho_{t_{r+1}}$ and $\text{ERC}(\log(2^{s+1}h_{t_{r+1}}))$.

The DECODE function (for Algorithm 1) simply reads $b$ bits to reconstruct $\|\mathbf{v}\|$. Using $\text{ERC}^{-1}$, it decodes the index of the first nonzero coordinate, reads the bit indicating the sign, and then uses $\text{ERC}^{-1}$ again to determines the quantization level of this first nonzero coordinate. The process proceeds in rounds, mimicking the encoding process, finishing when all coordinates have been decoded.

Like Alistarh et al. (2017), we use Elias recursive coding (Elias, 1975, ERC) to encode positive integers. ERC is simple and has several desirable properties, including the property that the coding scheme assigns shorter codes to smaller values, which makes sense in our scheme as they are more likely to occur. Elias coding is a universal lossless integer coding scheme with a recursive encoding and decoding structure.

The Elias recursive coding scheme is summarized in Algorithm 2. For any positive integer $N$, the following results are known for ERC (Alistarh et al., 2017):

1. $|\text{ERC}(N)| \leq (1 + o(1)) \log N + 1$;
2. $\text{ERC}(N)$ can be encoded and decoded in time $O(|\text{ERC}(N)|)$;

3. Decoding can be done without knowledge of an upper bound on $N$.

## B  PROOF OF THEOREM 2 (VARIANCE BOUND)

We first find a simple expression of the variance of $Q_s(\mathbf{v})$ for every arbitrary quantization scheme in the following lemma:

**Lemma 1.** *Let $\mathbf{v} \in \mathbb{R}^d$, $\mathscr{L} = (l_0, l_1, \cdots, l_{s+1})$, and fix $s \geq 1$. The variance of $Q_s(\mathbf{v})$ for general sequence of quantization levels is given by*

$$\mathbb{E}[\|Q_s(\mathbf{v}) - \mathbf{v}\|^2] = \|\mathbf{v}\|^2 \sum_{i=1}^{d} \tau^2(r_i) p(r_i)\big(1 - p(r_i)\big) \tag{5}$$

*where $r_i = |v_i|/\|\mathbf{v}\|$ and $p(r), \tilde{s}(r), \tau(r)$ are defined in Section 3.1.*

*Proof.* Noting the random quantization is i.i.d over elements of a stochastic gradient, we can decompose $\mathbb{E}[\|Q_s(\mathbf{v}) - \mathbf{v}\|^2]$ as:

$$\mathbb{E}[\|Q_s(\mathbf{v}) - \mathbf{v}\|^2] = \sum_{i=1}^{d} \|\mathbf{v}\|^2 \sigma^2(r_i) \tag{6}$$

where $\sigma^2(r_i) = \mathbb{E}[(h_i(\mathbf{v}, s) - r_i)^2]$. Computing the variance of $h_i(\mathbf{v}, s)$, we can show that $\sigma^2(r_i) = \tau^2(r_i) p(r_i)\big(1 - p(r_i)\big)$. $\quad\square$

In the following, we consider NUQSGD algorithm with $\hat{\mathscr{L}} = (0, 1/2^s, \cdots, 2^{s-1}/2^s, 1)$ as the quantization levels. Then, $h_i(\mathbf{v}, s)$'s are defined in two cases based on which quantization interval $r_i$ falls into:

1) If $r_i \in [0, 2^{-s}]$, then

$$h_i(\mathbf{v}, s) = \begin{cases} 0 & \text{with probability } 1 - p_1(r_i, s); \\ 2^{-s} & \text{otherwise} \end{cases} \tag{7}$$

where $p_1(r, s) = 2^s r$.

2) If $r_i \in [2^{j-s}, 2^{j+1-s}]$ for $j = 0, \cdots, s-1$, then

$$h_i(\mathbf{v}, s) = \begin{cases} 2^{j-s} & \text{with probability } 1 - p_2(r_i, s); \\ 2^{j+1-s} & \text{otherwise} \end{cases} \tag{8}$$

where $p_2(r, s) = 2^{s-j} r - 1$. Note that $Q_s(\mathbf{0}) = \mathbf{0}$.

Let $\mathscr{S}_j$ denote the coordinates of vector $\mathbf{v}$ whose elements fall into the $(j+1)$-th bin, *i.e.*, $\mathscr{S}_0 \triangleq \{i : r_i \in [0, 2^{-s}]\}$ and $\mathscr{S}_{j+1} \triangleq \{i : r_i \in [2^{j-s}, 2^{j+1-s}]\}$ for $j = 0, \cdots, s-1$. Let $d_j \triangleq |\mathscr{S}_j|$. Applying the result of Lemma 1, we have

$$\mathbb{E}[\|Q_s(\mathbf{v}) - \mathbf{v}\|^2] = \|\mathbf{v}\|^2 \tau_0^2 \sum_{i \in \mathscr{S}_0} p_1(r_i, s)(1 - p_1(r_i, s))$$
$$+ \|\mathbf{v}\|^2 \sum_{j=0}^{s-1} \tau_{j+1}^2 \sum_{i \in \mathscr{S}_{j+1}} p_2(r_i, s)\big(1 - p_2(r_i, s)\big) \tag{9}$$

where $\tau_j \triangleq l_{j+1} - l_j$ for $j \in \{0, \cdots, s\}$.

Substituting $\tau_0 = 2^{-s}$ and $\tau_j = 2^{j-1-s}$ for $j \in \{1, \cdots, s\}$ into (9), we have

$$\mathbb{E}[\|Q_s(\mathbf{v}) - \mathbf{v}\|^2] = \|\mathbf{v}\|^2 2^{-2s} \sum_{i \in \mathscr{S}_0} p_1(r_i, s)(1 - p_1(r_i, s))$$

$$+ \|\mathbf{v}\|^2 \sum_{j=0}^{s-1} 2^{2(j-s)} \sum_{i \in \mathscr{S}_{j+1}} p_2(r_i, s)\big(1 - p_2(r_i, s)\big)$$

$$\leq \|\mathbf{v}\|^2 2^{-2s} \sum_{i \in \mathscr{S}_0} p_1(r_i, s)$$

$$+ \|\mathbf{v}\|^2 \sum_{j=0}^{s-1} 2^{2(j-s)} \sum_{i \in \mathscr{S}_{j+1}} p_2(r_i, s) \tag{10}$$

We first note that $\sum_{i \in \mathscr{S}_0} p_1(r_i, s) \leq d$ and $\sum_{i \in \mathscr{S}_{j+1}} p_2(r_i, s) \leq d$ for all $j$, *i.e.,* an upper bound on the variance of $Q_s(\mathbf{v})$ is given by $\mathbb{E}[\|Q_s(\mathbf{v}) - \mathbf{v}\|^2] \leq \|\mathbf{v}\|^2 d/3(2^{-2s+1} + 1)$. Furthermore, we have

$$\sum_{i \in \mathscr{S}_0} p_1(r_i, s) \leq \min\{d_0, 2^s \sqrt{d_0}\} \tag{11}$$

since $\frac{\sum_{i \in \mathscr{S}_0} |v_i|}{\|\mathbf{v}\|} \leq \sqrt{d_0}$. Similarly, we have

$$\sum_{i \in \mathscr{S}_{j+1}} p_2(r_i, s) \leq \min\{d_{j+1}, 2^{(s-j)}\sqrt{d_{j+1}}\}. \tag{12}$$

Substituting the upper bounds in (11) and (12) into (10), an upper bound on the variance of $Q_s(\mathbf{v})$ is given by

$$\mathbb{E}[\|Q_s(\mathbf{v}) - \mathbf{v}\|^2] \leq \min\{2^{-2s}d_0, 2^{-s}\sqrt{d_0}\}\|\mathbf{v}\|^2$$

$$+ \sum_{j=0}^{s-1} \min\{2^{2(j-s)}d_{j+1}, 2^{j-s}\sqrt{d_{j+1}}\}\|\mathbf{v}\|^2. \tag{13}$$

The upper bound in (13) cannot be used directly as it depends on $\{d_0, \cdots, d_s\}$. Note that $d_j$'s depend on quantization intervals. In the following, we obtain an upper bound on $\mathbb{E}[\|Q_s(\mathbf{v}) - \mathbf{v}\|^2]$, which depends only on $d$ and $s$. To do so, we need to use this lemma inspired by (Alistarh et al., 2017, Lemma A.5): Let $\|\cdot\|_0$ count the number of nonzero components.

**Lemma 2.** *Let $\mathbf{v} \in \mathbb{R}^d$. The expected number of nonzeros in $Q_s(\mathbf{v})$ is bounded above by*

$$\mathbb{E}[\|Q_s(\mathbf{v})\|_0] \leq 2^{2s} + \sqrt{d_0}2^s.$$

*Proof.* Note that $d - d_0 \leq 2^{2s}$ since

$$(d - d_0)2^{-2s} \leq \sum_{i \notin \mathscr{S}_0} r_i^2 \leq 1. \tag{14}$$

For each $i \in \mathscr{S}_0$, $Q_s(v_i)$ becomes zero with probability $1 - 2^s r_i$, which results in

$$\mathbb{E}[\|Q_s(\mathbf{v})\|_0] \leq d - d_0 + \sum_{i \in \mathscr{S}_0} r_i 2^s$$

$$\leq 2^{2s} + \sqrt{d_0}2^s. \tag{15}$$

$\square$

Using a similar argument as in the proof of Lemma 2, we have

$$d - d_0 - d_1 - \cdots - d_j \leq 2^{2(s-j)} \tag{16}$$

for $j = 0, 1, \cdots, s - 1$. Define $b_j \triangleq d - 2^{2(s-j)}$ for $j = 0, \cdots, s - 1$. Then

$$b_0 \leq d_0$$
$$b_1 \leq d_1 + d_0$$
$$\vdots \quad \vdots$$
$$b_{s-1} \leq d_0 + \cdots + d_{s-1}. \tag{17}$$

Note that $d_s = d - d_0 - \cdots - d_{s-1}$.

We define

$$
\begin{aligned}
\tilde{d}_0 &\triangleq b_0 = d - 2^{2s} \\
\tilde{d}_1 &\triangleq b_1 - b_0 = 3 \cdot 2^{2(s-1)} \\
&\vdots \qquad \vdots \\
\tilde{d}_{s-1} &\triangleq b_{s-1} - b_{s-2} = 12 \\
\tilde{d}_s &\triangleq d - \tilde{d}_0 - \tilde{d}_1 - \cdots - \tilde{d}_{s-1} = 4.
\end{aligned}
\tag{18}
$$

Note that $\tilde{d}_0 \le d_0$, $\tilde{d}_1 + \tilde{d}_0 \le d_1 + d_0$, $\cdots$, $\tilde{d}_{s-1} + \cdots + \tilde{d}_0 \le d_{s-1} + \cdots + d_0$, and $\tilde{d}_s + \cdots + \tilde{d}_0 = d_s + \cdots + d_0$.

Noting that the coefficients of the additive terms in the upper bound in (13) are monotonically increasing with $j$, we can find an upper bound on $\mathbb{E}[\|Q_s(\mathbf{v}) - \mathbf{v}\|^2]$ by replacing $(d_0, \cdots, d_s)$ with $(\tilde{d}_0, \cdots, \tilde{d}_s)$ in (13), which gives (3) and completes the proof.

## C    PROOF OF THEOREM 3 (CODE-LENGTH BOUND)

Let $|\cdot|$ denote the length of a binary string. In this section, we find an upper bound on $\mathbb{E}[|\text{ENCODE}(\mathbf{v})|]$, *i.e.*, the expected number of communication bits per iteration. Recall from Appendix A that the quantized gradient $Q_s(\mathbf{v})$ is determined by the tuple $(\|\mathbf{v}\|, \boldsymbol{\rho}, \mathbf{h})$. Write $i_1 < i_2 < \cdots < i_{\|\mathbf{h}\|_0}$ for the indices of the $\|\mathbf{h}\|_0$ nonzero entries of $\mathbf{h}$. Let $i_0 = 0$.

The encoding produced by $\text{ENCODE}(\mathbf{v})$ can be partitioned into two parts, $R$ and $E$, such that, for $j = 1, \ldots, \|\mathbf{h}\|_0$,

- $R$ contains the codewords $\text{ERC}(i_j - i_{j-1})$ encoding the runs of zeros; and

- $E$ contains the sign bits and codewords $\text{ERC}(\log\{2^{s+1}h_{i_j}\})$ encoding the normalized quantized coordinates.

Note that $\|[i_1, i_2 - i_1, \cdots, i_{\|\mathbf{h}\|_0} - i_{\|\mathbf{h}\|_0 - 1}]\|_1 \le d$. Thus, by (Alistarh et al., 2017, Lemma A.3), the properties of Elias encoding imply that

$$
|R| \le \|\mathbf{h}\|_0 + (1 + o(1))\|\mathbf{h}\|_0 \log\left(\frac{d}{\|\mathbf{h}\|_0}\right).
\tag{19}
$$

We now turn to bounding $|E|$. The following result in inspired by (Alistarh et al., 2017, Lemma A.3).

**Lemma 3.** *Fix a vector $\mathbf{q}$ such that $\|\mathbf{q}\|_p^p \le P$, let $i_1 < i_2 < \ldots i_{\|\mathbf{q}\|_0}$ be the indices of its $\|\mathbf{q}\|_0$ nonzero entries, and assume each nonzero entry is of form of $2^k$, for some positive integer $k$. Then*

$$
\sum_{j=1}^{\|\mathbf{q}\|_0} |\text{ERC}(\log(q_{i_j}))| \le (1 + o(1))\log\left(\frac{1}{p}\right) + \|\mathbf{q}\|_0
$$

$$
+ (1 + o(1))\|\mathbf{q}\|_0 \log\log\left(\frac{P}{\|\mathbf{q}\|_0}\right).
$$

*Proof.* Applying property (1) for ERC (end of Appendix A), we have

$$\sum_{j=1}^{\|\mathbf{q}\|_0} |\mathrm{ERC}(\log(q_{i_j}))| \leq (1+o(1)) \sum_{j=1}^{\|\mathbf{q}\|_0} \log\log q_{i_j} + \|\mathbf{q}\|_0$$

$$\leq (1+o(1))\log\left(\frac{1}{p}\right) + \|\mathbf{q}\|_0$$

$$+ (1+o(1)) \sum_{j=1}^{\|\mathbf{q}\|_0} \log\log q_{i_j}^p$$

$$\leq (1+o(1))\log\left(\frac{1}{p}\right) + \|\mathbf{q}\|_0$$

$$+ (1+o(1))\|\mathbf{q}\|_0 \log\log\left(\frac{P}{\|\mathbf{q}\|_0}\right)$$

where the last bound is obtained by Jensen's inequality. □

Taking $\mathbf{q} = 2^{s+1}\mathbf{h}$, we note that $\|\mathbf{q}\|^2 = 2^{2s+2}\|\mathbf{h}\|^2$ and

$$\|\mathbf{h}\|^2 \leq \sum_{i=1}^{d} \left(\frac{v_i}{\|\mathbf{v}\|} + \frac{1}{2^s}\right)^2$$

$$\leq 2\sum_{i=1}^{d} \left(\frac{v_i^2}{\|\mathbf{v}\|^2} + \frac{1}{2^{2s}}\right) = 2\left(1 + \frac{d}{2^{2s}}\right). \tag{20}$$

By Lemma 3 applied to $\mathbf{q}$ and the upper bound (20),

$$|E| \leq -(1+o(1)) + 2\|\mathbf{h}\|_0$$

$$+ (1+o(1))\|\mathbf{h}\|_0 \log\log\left(\frac{2^{2s+2}\|\mathbf{h}\|^2}{\|\mathbf{h}\|_0}\right). \tag{21}$$

Combining (19) and (21), we obtain an upper bound on the expected code-length:

$$\mathbb{E}[|\mathrm{ENCODE}(\mathbf{v})|] \leq N(\|\mathbf{h}\|_0) \tag{22}$$

where

$$N(\|\mathbf{h}\|_0) = b + 3\|\mathbf{h}\|_0 + (1+o(1))\mathbb{E}\left[\|\mathbf{h}\|_0 \log\left(\frac{d}{\|\mathbf{h}\|_0}\right)\right]$$

$$- (1+o(1)) + (1+o(1))\mathbb{E}\left[\|\mathbf{h}\|_0 \log\log\left(\frac{8(2^{2s}+d)}{\|\mathbf{h}\|_0}\right)\right]. \tag{23}$$

It is not difficult to show that, for all $k > 0$, $g_1(x) \triangleq x\log\left(\frac{k}{x}\right)$ is concave. Note that $g_1$ is an increasing function up to $x = k/e$.

Defining $g_2(x) \triangleq x\log\log\left(\frac{C}{x}\right)$ and taking the second derivative, we have

$$g_2''(x) = -\left(x\ln(2)\ln(C/x)\right)^{-1}\left(1 + \left(\ln(C/x)\right)^{-1}\right). \tag{24}$$

Hence $g_2$ is also concave on $x < C$. Furthermore, $g_2$ is increasing up to some $C/5 < x^* < C/4$. We note that $\mathbb{E}[\|\mathbf{h}\|_0] \leq 2^{2s} + \sqrt{d}2^s$ following Lemma 2. By assumption $2^{2s} + \sqrt{d}2^s \leq d/e$, and so, Jensen's inequality and (22) lead us to (4).

## D    PROOF OF THEOREM 4 (NUQSGD FOR SMOOTH CONVEX OPTIMIZATION)

Let $g(\mathbf{w})$ and $\hat{g}(\mathbf{w})$ denote the full-precision and decoded stochastic gradients, respectively. Then

$$\mathbb{E}[\|\hat{g}(\mathbf{w}) - \nabla f(\mathbf{w})\|^2] \leq \mathbb{E}[\|g(\mathbf{w}) - \nabla f(\mathbf{w})\|^2] + \mathbb{E}[\|\hat{g}(\mathbf{w}) - g(\mathbf{w})\|^2]. \tag{25}$$

By Theorem 2, $\mathbb{E}[\|\hat{g}(\mathbf{w}) - g(\mathbf{w})\|^2] \leq \varepsilon_Q \mathbb{E}[\|g(\mathbf{w})\|^2]$. By assumption, $\mathbb{E}[\|g(\mathbf{w})\|^2] \leq B$. Noting $g(\mathbf{w})$ is unbiased, $\mathbb{E}[\|\hat{g}(\mathbf{w}) - \nabla f(\mathbf{w})\|^2] \leq (1+\varepsilon_Q)B$. The result follows by Corollary 1.

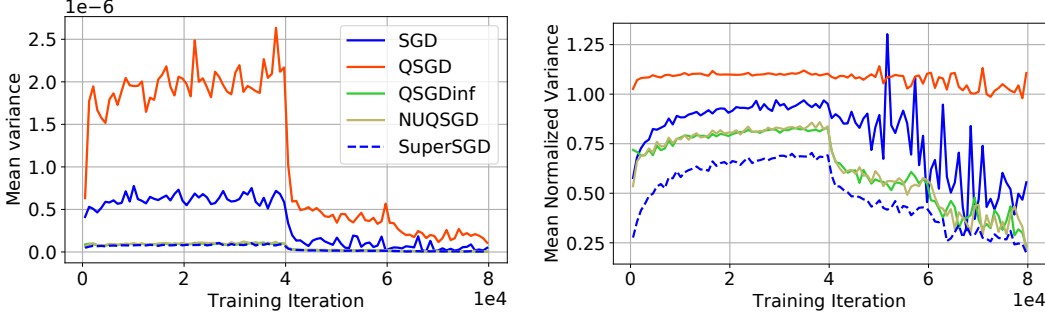

Figure 5: Estimated variance (left) and normalized variance (right) on CIFAR10 on the trajectory of single-GPU SGD. Variance is measured for fixed model snapshots during training. Notice that the variance for NUQSGD and QSGDinf is lower than SGD for almost all the training and it decreases after the learning rate drops. All methods except SGD simulate training using 8 GPUs. SuperSGD applies no quantization to the gradients and represents the lowest variance we could hope to achieve.

## E  PROOF OF THEOREM 5 (EXPECTED NUMBER OF COMMUNICATION BITS)

Assuming $\frac{2\hat{B}}{K\varepsilon^2} > \frac{\beta}{\varepsilon}$, then $T_\varepsilon = O\left(\frac{2\hat{B}}{K\varepsilon^2}R^2\right)$. Ignoring all but terms depending on $d$ and $s$, we have $T_\varepsilon = O(\hat{B}/\varepsilon^2)$. Following Theorems 2 and 3 for NUQSGD, $\zeta_{\text{NUQSGD},\varepsilon} = O(N_Q\varepsilon_Q B/\varepsilon^2)$. For QSGD, following the results of Alistarh et al. (2017), $\zeta_{\text{QSGD},\varepsilon} = O(\tilde{N}_Q\tilde{\varepsilon}_Q B/\varepsilon^2)$ where $\tilde{N}_Q = 3(s^2 + s\sqrt{d}) + (\frac{3}{2} + o(1))(s^2 + s\sqrt{d})\log\left(\frac{2(s^2+d)}{s^2+\sqrt{d}}\right) + b$ and $\tilde{\varepsilon}_Q = \min\left(\frac{d}{s^2}, \frac{\sqrt{d}}{s}\right)$.

In overparameterized networks, where $d \geq 2^{2s+1}$, we have $\varepsilon_Q = 2^{-s}\sqrt{d - 2^{2s}} + O(s)$ and $\tilde{\varepsilon}_Q = \sqrt{d}/s$. Furthermore, for sufficiently large $d$, $N_Q$ and $\tilde{N}_Q$ are given by $O\left(2^s\sqrt{d}\log\left(\frac{\sqrt{d}}{2^s}\right)\right)$ and $O\left(s\sqrt{d}\log(\sqrt{d})\right)$, respectively.

## F  ADDITIONAL EXPERIMENTS

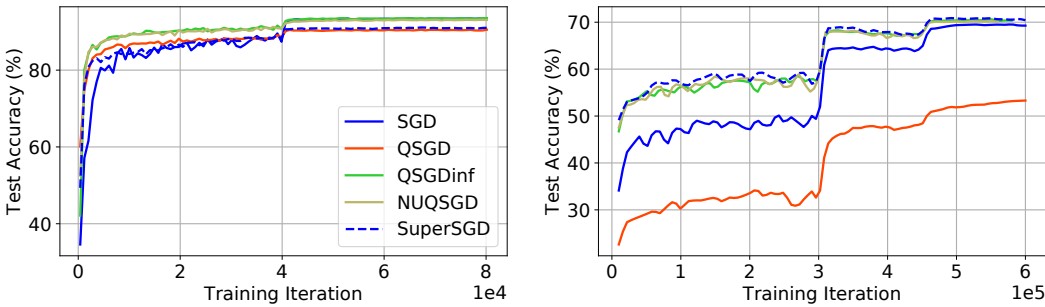

Figure 6: Accuracy on the hold-out set on CIFAR10 (left) and on ImageNet (right) for training ResNet models from random initialization until convergence. For CIFAR10, the hold-out set is the test set and for ImageNet, the hold-out set is the validation set.

In this section, we present further experimental results in a similar setting to Section 5.

In Figure 6, we show the test accuracy for training ResNet110 on CIFAR10 and validation accuracy for training ResNet34 on ImageNet from random initialization until convergence (discussed in Section 5). Similar to the training loss performance, we observe that NUQSGD and QSGDinf outperform QSGD in terms of test accuracy in both experiments. In both experiments, unlike NUQSGD, QSGD does not recover the test accuracy of SGD. The gap between NUQSGD and QSGD on ImageNet is significant. We argue that this is achieved because NUQSGD and QSGDinf have lower variance relative to QSGD. It turns out both training loss and generalization error can benefit from the reduced variance.

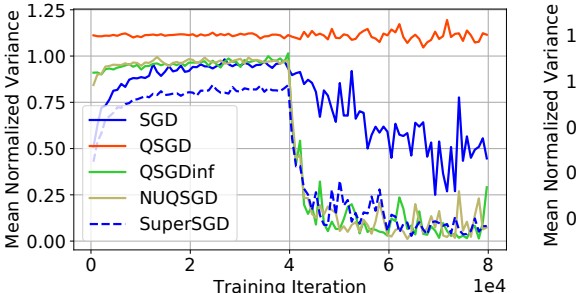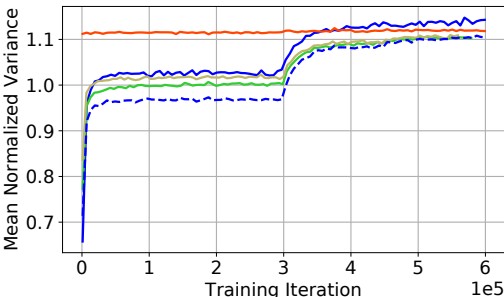

Figure 7: Estimated normalized variance on CIFAR10 (left) and ImageNet (right). For different methods, the variance is measured on their own trajectories. Note that the normalized variance of NUQSGD and QSGDinf is lower than SGD for almost the entire training. It decreases on CIFAR10 after the learning rate drops and does not grow as much as SGD on ImageNet. Since the variance depends on the optimization trajectory, these curves are not directly comparable. Rather the general trend should be studied.

We also measure the variance and normalized variance at fixed snapshots during training by evaluating multiple gradient estimates using each quantization method. All methods are evaluated on the same trajectory traversed by the single-GPU SGD. These plots answer this specific question: What would the variance of the first gradient estimate be if one were to train using SGD for any number of iterations then continue the optimization using another method? The entire future trajectory may change by taking a single good or bad step. We can study the variance along any trajectory. However, the trajectory of SGD is particularly interesting because it covers a subset of points in the parameter space that is likely to be traversed by any first-order optimizer. For multi-dimensional parameter space, we average the variance of each dimension.

Figure 5 (left), shows the variance of the gradient estimates on the trajectory of single-GPU SGD on CIFAR10. We observe that QSGD has particularly high variance, while QSGDinf and NUQSGD have lower variance than single-GPU SGD.

We also propose another measure of stochasticity, normalized variance, that is the variance normalized by the norm of the gradient. The mean normalized variance can be expressed as

$$\frac{\mathbb{E}_i[V_A[\partial l(\mathbf{w};\mathbf{z})/\partial w_i]]}{\mathbb{E}_i[\mathbb{E}_A[(\partial l(\mathbf{w};\mathbf{z})/\partial w_i)^2]]}$$

where $l(\mathbf{w};\mathbf{z})$ denotes the loss of the model parametrized by $\mathbf{w}$ on sample $\mathbf{z}$ and subscript $A$ refers to randomness in the algorithm, *e.g.,* randomness in sampling and quantization. Normalized variance can be interpreted as the inverse of Signal to Noise Ratio (SNR) for each dimension. We argue that the noise in optimization is more troubling when it is significantly larger than the gradient. For sources of noise such as quantization that stay constant during training, their negative impact might only be observed when the norm of the gradient becomes small.

Figure 5 (right) shows the mean normalized variance of the gradient versus training iteration. Observe that the normalized variance for QSGD stays relatively constant while the unnormalized variance of QSGD drops after the learning rate drops. It shows that the quantization noise of QSGD can cause slower convergence at the end of the training than at the beginning.

In Figure 7, we show the mean normalized variance of the gradient versus training iteration on CIFAR10 and ImageNet. For different methods, the variance is measured on their own trajectories. Since the variance depends on the optimization trajectory, these curves are not directly comparable. Rather the general trend should be studied.

**ResNet152 Weak Scaling.** In Figure 8, we present the weak scaling results for ResNet152/ImageNet. Each of the GPUs receives a batch of size 8, and we therefore scale up the global batch size by the number of nodes. The results exhibit the same superior scaling behavior for NUQSGD relative to the uncompressed baseline.

**EF-SignSGD Convergence.** In Figure 9, we present a performance comparison for NUQSGD variants (bucket size 512) and a convergent variant of EF-SignSGD with significant levels of parameter

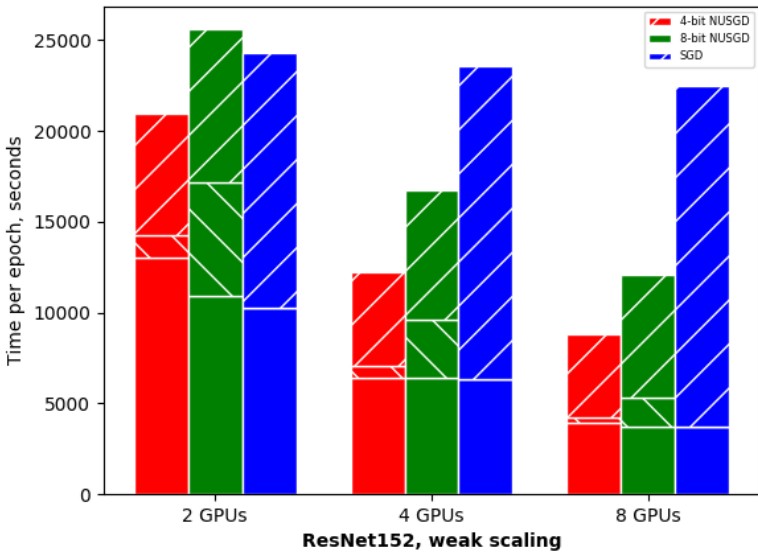

Figure 8: Scalability behavior for NUQSGD versus the full-precision baseline when training ResNet152 on ImageNet.

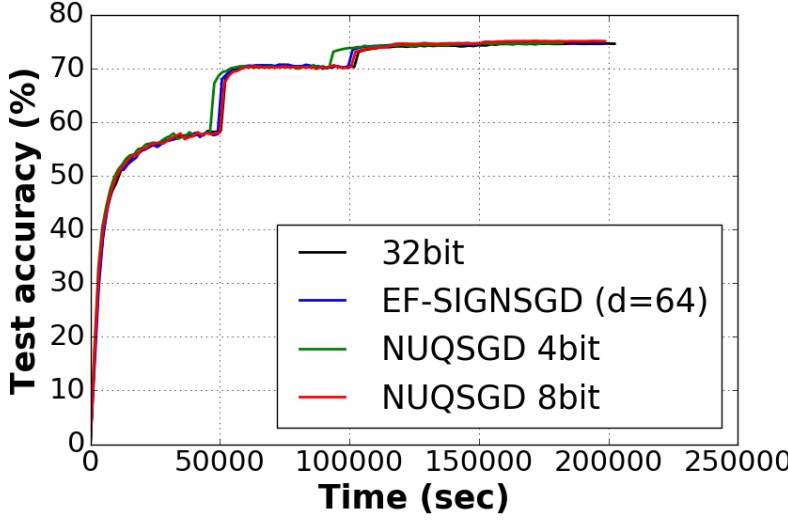

Figure 9: End-to-end training time for ResNet50/ImageNet for NUQSGD and EF-SignSGD versus the SGD baseline.

tuning for convergence. We believe this to be the first experiment to show convergence of the latter method at ImageNet scale, as the original paper only considers the CIFAR dataset. For convergence, we have tuned the choice of scaling factor and the granularity at which quantization is applied (bucket size). We have also considered learning rate tuning, but that did not appear to prevent divergence in the early stages of training for this model. We did not attempt warm start, since that would significantly decrease the practicality of the algorithm. We have found that bucket size 64 is the highest at which the algorithm will still converge on this model and dataset, and found 1-bit SGD scaling (Seide et al., 2014), which consists of taking sums over positives and over negatives for each bucket, to yield good results. The experiments are executed on a machine with 8 NVIDIA Titan X

GPUs, and batch size 256, and can be found in Figure 9. Under these hyperparameter values the EF-SignSGD algorithm sends 128 bits per each bucket of 64 values (32 for each scaling factor, and 64 for the signs), doubling its baseline communication cost. Moreover, the GPU implementation is not as efficient, as error feedback must be computed and updated at every step, and there is less parallelism to leverage inside each bucket. This explains the fact that the end-to-end performance is in fact close to that of the 8-bit NUQSGD variant, and inferior to 4-bit NUQSGD.

## G   VARIANCE LOWER BOUND

In the following theorem, we show that for any given set of levels, there exists a distribution of points with dimension d such that the variance is in $\Omega(\sqrt{d})$, and so our bound is tight in $d$.

**Theorem 6** (Lower bound). *Let $d \in \mathbb{Z}^{>0}$ and let $(0, l_1, \cdots, l_s, 1)$ denote an arbitrary sequence of quantization levels. Provided $d \geq (2/l_1)^2$, there exists a vector $\mathbf{v} \in \mathbb{R}^d$ such that the variance of unbiased quantization of $\mathbf{v}$ is lower bounded by $\|\mathbf{v}\|^2 l_1 \sqrt{d}/2$, i.e., the variance is in $\Omega(\sqrt{d})$.*

*Proof.* The variance of $Q_s(\mathbf{v})$ for general sequence of quantization levels is given by

$$\mathbb{E}[\|Q_s(\mathbf{v}) - \mathbf{v}\|^2] = \|\mathbf{v}\|^2 \sum_{i=1}^{d} \sigma^2(r_i).$$

If $r \in [l_{\tilde{s}(r)}, l_{\tilde{s}(r)+1}]$, the variance $\sigma^2(r)$ can be expressed as

$$\sigma^2(r) = \tau(r)^2 p(r)(1 - p(r)) = (l_{\tilde{s}(r)+1} - r)(r - l_{\tilde{s}(r)}). \tag{26}$$

We consider $\mathbf{v}_0 = [r, r, \cdots, r]^T$ for $r \neq 0$. The normalized coordinates is $\hat{\mathbf{v}}_0 = [1/\sqrt{d}, \cdots, 1/\sqrt{d}]^T$.

Using (26) and noting $1/\sqrt{d} < l_1$, we have

$$\sigma^2(r_0) = 1/\sqrt{d}\left(l_1 - 1/\sqrt{d}\right) \geq l_1/(2\sqrt{d}). \tag{27}$$

Summing variance of all coordinates and applying (27), the variance of $Q_s(\mathbf{v}_0)$ is lower bounded by

$$\mathbb{E}[\|Q_s(\mathbf{v}_0) - \mathbf{v}_0\|^2] = \|\mathbf{v}_0\|^2 d\sigma^2(r) \geq \|\mathbf{v}_0\|^2 l_1 \sqrt{d}/2. \tag{28}$$

## H   WORST-CASE VARIANCE ANALYSIS

In this section, we first derive the optimal worst-case variance upper bound by optimizing over the distribution of normalized coordinates for an arbitrary sequence of levels, expressed as a solution to an integer program with quadratic constraints. We then relax the program to obtain a quadratically constrained quadratic program (QCQP). A coarser analysis yields an upper bound expressed as a solution to a linear program (LP), which is more amenable to analysis. We solve this LP analytically for the special case of $s = 1$ and show the optimal level is at $1/2$.

Then, for an exponentially spaced collection of levels of the form $(0, p^s, \cdots, p^2, p, 1)$ for $p \in [0, 1]$ and an integer number of levels, $s$, we write the expression of QCQP and solve it efficiently using standard solvers. We have a numerical method for finding the optimal value of $p$ that minimizes the worst-case variance, for any given $s$ and $d$. Through the worst-case analysis, we gain insight into the behaviour of the variance upper bound. We show that our current scheme is nearly optimal (in the worst-case sense) in some cases. Using these techniques we can obtain slightly tighter bounds numerically.

### H.1   GENERALLY SPACED LEVELS

Let $\mathscr{L} = (l_0, l_1, \cdots, l_s, l_{s+1})$ denote an arbitrary sequence of quantization levels where $l_0 = 0 < l_1 < \cdots < l_{s+1} = 1$. Recall that, for $r \in [0, 1]$, we define $\tilde{s}(r)$ and $p(r)$ such that they satisfy $l_{\tilde{s}(r)} \leq r \leq l_{\tilde{s}(r)+1}$ and $r = (1 - p(r))l_{\tilde{s}(r)} + p(r)l_{\tilde{s}(r)+1}$, respectively. Define $\tau(r) = l_{\tilde{s}(r)+1} - l_{\tilde{s}(r)}$. Note that

$\tilde{s}(r) \in \{0, 1, \cdots, s\}$. Then, $h_i(\mathbf{v}, s)$'s are defined in two cases based on which quantization interval $r_i$ falls into:

1) If $r_i \in [0, l_1]$, then

$$h_i(\mathbf{v}, s) = \begin{cases} 0 & \text{with probability } 1 - p_1(r_i, \mathscr{L}); \\ l_1 & \text{otherwise} \end{cases} \tag{29}$$

where $p_1(r, \mathscr{L}) = r/l_1$.

2) If $r_i \in [l_{j-1}, l_j]$ for $j = 1, \cdots, s+1$, then

$$h_i(\mathbf{v}, s) = \begin{cases} l_{j-1} & \text{with probability } 1 - p_2(r_i, \mathscr{L}); \\ l_j & \text{otherwise} \end{cases} \tag{30}$$

where $p_2(r, \mathscr{L}) = (r - l_{j-1})/\tau_{j-1}$.

Let $\mathscr{S}_j$ denote the coordinates of vector $\mathbf{v}$ whose elements fall into the $(j+1)$-th bin, *i.e.*, $\mathscr{S}_j \triangleq \{i : r_i \in [l_j, l_{j+1}]\}$ for $j = 0, \cdots, s$. Let $d_j \triangleq |\mathscr{S}_j|$.

Following Lemma 1 and steps in Theorem 2, we can show that

$$\mathbb{E}[\|Q_s(\mathbf{v}) - \mathbf{v}\|^2] \leq \|\mathbf{v}\|^2 \sum_{j=0}^{s} \min\{\tau_j^2 d_j, \tau_j \sqrt{d_j}\}. \tag{31}$$

**Theorem 7** (QCQP bound). *Let $\mathbf{v} \in \mathbb{R}^d$. An upper bound on the nonuniform quantization of $\mathbf{v}$ is given by $\varepsilon_{QP}\|\mathbf{v}\|^2$ where $\varepsilon_{QP}$ is the optimal value of the following QCQP:*

$$\mathscr{Q}_1 : \quad \max_{(d_0, \cdots, d_s, z_0, \cdots, z_s)} \sum_{j=0}^{s} z_j$$

$$\text{subject to } d - d_0 - \cdots - d_j \leq (1/l_{j+1})^2, \; j = 0, \cdots, s-1,$$

$$\sum_{j=0}^{s} d_j \leq d,$$

$$z_j \leq \tau_j^2 d_j, \; z_j^2 \leq \tau_j^2 d_j, \; j = 0, \cdots, s,$$

$$d_j \geq 0, \; j = 0, \cdots, s.$$

*Proof.* Following Lemma 2, we have

$$d - d_0 - d_1 - \cdots - d_j \leq (1/l_{j+1})^2 \tag{32}$$

for $j = 0, \cdots, s-1$.

The problem of optimizing $(d_0, \cdots, d_s)$ to maximize the variance upper bound (31) subject to (32) is given by

$$\mathscr{R}_1 : \quad \max_{(d_0, \cdots, d_s)} \sum_{j=0}^{s} \min\{\tau_j^2 d_j, \tau_j \sqrt{d_j}\}$$

$$\text{subject to (32), } j = 0, \cdots, s-1,$$

$$\sum_{j=0}^{s} d_j \leq d, \tag{33}$$

$$d_j \in \mathbb{Z}^{\geq 0}, \; j = 0, \cdots, s. \tag{34}$$

Let $z_j \triangleq \min\{\tau_j^2 d_j, \tau_j \sqrt{d_j}\}$ denote an auxiliary variable for $j = 0, \cdots, s$. Problem $\mathscr{R}_1$ can be rewritten as

$$\mathscr{R}_2 : \quad \max_{(d_0, \cdots, d_s, z_0, \cdots, z_s)} \sum_{j=0}^{s} z_j$$

$$\text{subject to } z_j \leq \tau_j^2 d_j, \; z_j^2 \leq \tau_j^2 d_j, \; j = 0, \cdots, s, \tag{35}$$

$$(32), (33), \text{ and } (34).$$

The variance optimization problem $\mathcal{R}_2$ is an integer nonconvex problem. We can obtain an upper bound on the optimal objective of problem $\mathcal{R}_2$ by relaxing the integer constraint as follows. The resulting QSQP is shown as follows:

$$\mathcal{Q}_1 : \quad \max_{(d_0,\cdots,d_s,z_0,\cdots,z_s)} \sum_{j=0}^{s} z_j$$
$$\text{subject to } d_j \geq 0, \ j = 0, \cdots, s, \tag{36}$$
$$(32), \ (33), \text{ and } (35).$$

Note that problem $\mathcal{Q}_1$ can be solved efficiently using standard standard interior point-based solvers, *e.g.,* CVX (Boyd and Vandenberghe, 2004). $\qquad\square$

In the following, we develop a coarser analysis that yields an upper bound expressed as the optimal value to an LP.

**Theorem 8** (LP bound). *Let $\mathbf{v} \in \mathbb{R}^d$. An upper bound on the nonuniform quantization of $\mathbf{v}$ is given by $\varepsilon_{LP}\|\mathbf{v}\|^2$ where $\varepsilon_{LP}$ is the optimal value of the following LP:*

$$\mathcal{P}_1 : \quad \max_{(d_0,\cdots,d_s)} \sum_{j=0}^{s} \tau_j^2 d_j$$
$$\text{subject to } d - d_0 - \cdots - d_j \leq (1/l_{j+1})^2, \ j = 0, \cdots, s-1,$$
$$\sum_{j=0}^{s} d_j \leq d,$$
$$d_j \geq 0, \ j = 0, \cdots, s.$$

*Proof.* The proof follows the steps in the proof of Theorem 7 for the problem of optimizing $(d_0, \cdots, d_s)$ to maximize the following upper bound

$$\mathbb{E}[\|Q_s(\mathbf{v}) - \mathbf{v}\|^2] \leq \|\mathbf{v}\|^2 \sum_{j=0}^{s} \tau_j^2 d_j. \tag{37}$$

$\qquad\square$

**Corollary 2** (Optimal level). *For the special case with $s = 1$, the optimal level to minimize the worst-case bound obtained from problem $\mathcal{P}_1$ is given by $l_1^* = 1/2$.*

*Proof.* For $s = 1$, problem $\mathcal{P}_1$ is given by

$$\mathcal{P}_0 : \max_{(d_0,d_1)} \tau_0^2 d_0 + \tau_1^2 d_1$$
$$\text{subject to } d - d_0 \leq (1/l_1)^2,$$
$$d_0 + d_1 \leq d,$$
$$d_0 \geq 0, \ d_1 \geq 0.$$

Note that the objective of $\mathcal{P}_0$ is monotonically increasing in $(d_0, d_1)$. It is not difficult to verify that the optimal $(d_0^*, d_1^*)$ is a corner point on the boundary line of the feasibility region of $\mathcal{P}_0$. Geometrical representation shows that that candidates for an optimal solution are $(d - (1/l_1)^2, (1/l_1)^2)$ and $(d, 0)$. Substituting into the objective of $\mathcal{P}_0$, the optimal value of $\mathcal{P}_0$ is given by

$$\varepsilon_{LP}^* = \max\{\tau_0^2 d, \tau_0^2 d + \tau_1^2/\tau_0^2 - 1\}. \tag{38}$$

Finally, note that $\tau_0 = \tau_1 = 1/2$ minimizes the optimal value of $\mathcal{P}_0$ (38). $\qquad\square$

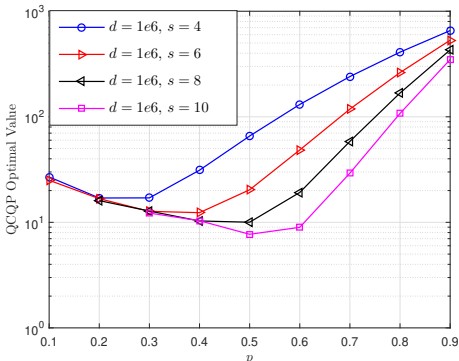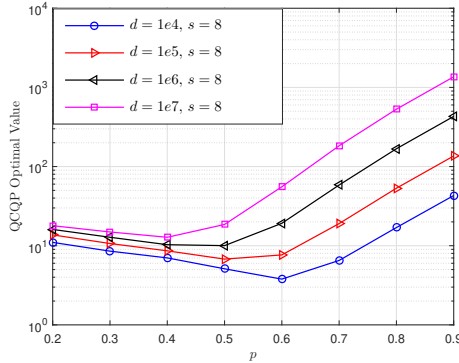

Figure 10: Optimal value of problem $\mathcal{Q}_1$ versus $p \in [0,1]$ for exponentially spaced collection of levels of the form $(0, p^s, \cdots, p^2, p, 1)$.

## H.2 Exponentially Spaced Levels

In this section, we focus on the special case of exponentially spaced collection of levels of the form $\mathscr{L}_p = (0, p^s, \cdots, p^2, p, 1)$ for $p \in [0,1]$ and an integer number of levels, $s$. In this case, we have $\tau_0 = p^s$ and $\tau_j = (1-p)p^{s-j}$ for $j = 1, \cdots, s$.

For any given $s$ and $d$, we can solve the corresponding quadratic and linear programs efficiently to find the worst-case variance bound. As a bonus, we can find the optimal value of $p$ that minimizes the worst-case variance bound. In Figure 10, we show the numerical results obtained by solving QCQP $\mathcal{Q}_1$ with $\mathscr{L}_p$ versus $p$ using CVX (Boyd and Vandenberghe, 2004). In Figure 10 (left), we fix $d$ and vary $s$, while in Figure 10 (right), we fix $s$ and vary $d$. As expected, we note that the variance upper bound increases as $d$ increases and the variance upper bound decreases as $s$ increases. We observe that our current scheme is nearly optimal (in the worst-case sense) in some cases. Further, the optimal value of $p$ shifts to the right as $d$ increases and shifts to the left as $s$ increases.

## I Nonconvex optimization

We can obtain convergence guarantees to various learning problems where we have convergence guarantees for SGD under standard assumptions. On nonconvex problems, (weaker) convergence guarantees can be established along the lines of, e.g., (Ghadimi and Lan, 2013, Theorem 2.1). In particular, NUQSGD is guaranteed to converge to a local minima for smooth general loss functions.

**Theorem 9** (NUQSGD for smooth nonconvex optimization). *Let $f : \Omega \to \mathbb{R}$ denote a possibly nonconvex and $\beta$-smooth function. Let $\mathbf{w}_0 \in \Omega$ denote an initial point, $\varepsilon_Q$ be defined as in Theorem 2, $T \in \mathbb{Z}^{>0}$, and $f^* = \inf_{\mathbf{w} \in \Omega} f(\mathbf{w})$. Suppose that Algorithm 1 is executed for $T$ iterations with a learning rate $\alpha = O(1/\beta)$ on $K$ processors, each with access to independent stochastic gradients of $f$ with a second-moment bound $B$. Then there exists a random stopping time $R \in \{0, \cdots, T\}$ such that NUQSGD guarantees $\mathbb{E}[\|\nabla f(\mathbf{w}_R)\|^2] \leq \varepsilon$ where $\varepsilon = O\big(\beta(f(\mathbf{w}_0) - f^*)/T + (1 + \varepsilon_Q)B\big)$.*

$\square$

