# OpenReview forum: "Provably Communication-efficient Data-parallel SGD via Nonuniform Quantization"
_ICLR.cc/2020/Conference — Reject_

### Official Review · AnonReviewer1 · 2019-10-23
**Official Blind Review #1**

**Rating:** 3

**Review:**

In this paper, the authors propose a new gradient compression method, which is called nonuniform quantization. The algorithm is a reasonable variant of SGD with uniform quantization. The paper is well written. The experiments show good performance.

However, there are several weakness in this paper:

1. In this paper, a very important reference and baseline is missing, which is call error-feedback SGD [1]. Although the title of [1] focuses on SignSGD, it provides a general algorithm for arbitrary compressor with a error/variance bound similar to Theorem 2 in this paper, no matter the compressor is unbiased or not. Since [1] provides the SOTA results for quantized SGD, the proposed algorithm should be compared to it in the experiments.

2. This paper claims to have strong theoretical guarantees. However, the theoretical analysis only works for convex functions. Note that the theoretical analysis in [1] also works for non-convex functions.

3. Regardless of the convergence guarantees (which is weak considering the existing theorems in [1]). the proposed algorithm, NUQSGD, does not show improvement on the convergence, compared to the baseline QSGDinf.

4. In Figure 3, the experiments only show loss vs. # of iterations, which does not show the actual training time. In Figure 4, training time is only shown for NUQSGD, which ignores the other baselines including QSGD and QSGDinf. What I really what to see is training loss (or testing accuracy) vs. training time (or communication overhead, such as number of bits), so that we can evaluate the trade-off between communication overhead and the convergence, compared to the baselines.



Minor issue (I hope the authors can consider the following suggestions in a revised version. However, since the issue is minor, it doesn't affect the score):

!. In Definition 1, in some cases $s$ is c constant integer, and in some other case $s$ become a function, which is very confusing and not friendly to the readers. I also hope the authors can highlight the definition of $r$ and $p$, which are essential for understanding the nonuniform quantization mechanism.




--------------
Reference

[1] Karimireddy, Sai Praneeth et al. “Error Feedback Fixes SignSGD and other Gradient Compression Schemes.” ICML (2019).

**Experience Assessment:**

I have read many papers in this area.

**Review Assessment: Checking Correctness Of Derivations And Theory:**

I assessed the sensibility of the derivations and theory.

**Review Assessment: Checking Correctness Of Experiments:**

I assessed the sensibility of the experiments.

**Review Assessment: Thoroughness In Paper Reading:**

I read the paper at least twice and used my best judgement in assessing the paper.

---

> ### Author Response · Authors · 2019-11-12
> **Response to Review 1**
>
> Thanks for your feedback. Below is our specific feedback to your review. We have also posted a general response (see our top-level comment) to all reviewers addressing high level points.
>
> >In this paper, a very important reference and baseline is missing, which is call error-feedback SGD [1]. Although the title of [1] focuses on SignSGD, it provides a general algorithm for arbitrary compressor with an error/variance bound similar to Theorem 2 in this paper, no matter the compressor is unbiased or not. Since [1] provides the SOTA results for quantized SGD, the proposed algorithm should be compared to it in the experiments.
>
> We agree that it is interesting to compare NUQSGD with error-corrected methods (although we feel this comparison is orthogonal to the problem of closing the performance--theory gap between QSGD and QSGDinf.) We are running experiments comparing NUQSGD with error-corrected methods and hope they are finished before the rebuttal period. Regardless, we will include them in the final paper. One important note is that error-corrected signSGD, sparsified methods, and TernGrad require non-trivial additional parameter tuning to reduce accuracy loss (learning rate. momentum, and warmup tuning---see e.g. "Deep Gradient Compression"). By contrast, our experiments target the setting where training is performed with standard hyperparameters as the full-precision version, and we are able to recover full accuracy in this regime. This is the standard set by QSGD, which is closer to practical applications.
>
> >This paper claims to have strong theoretical guarantees. However, the theoretical analysis only works for convex functions. Note that the theoretical analysis in [1] also works for non-convex functions.
>
> Using standard arguments, NUQSGD does provide guarantees in the non-convex case as well, since the quantized stochastic gradients are still unbiased. (This is the same for QSGD.) We mention this in the paper, after Theorem 4: "On nonconvex problems, convergence guarantees can be established along the lines of, e.g., (Ghadimi and Lan, 2013, Theorem 2.1)." In particular, this results gives convergence to a second-order stationary point. These are virtually the same guarantees as error-corrected signSGD.  We will include the nonconvex convergence statement in the updated paper.
>
> >Regardless of the convergence guarantees (which is weak considering the existing theorems in [1]). the proposed algorithm, NUQSGD, does not show improvement on the convergence, compared to the baseline QSGDinf.
>
> The goal of the paper was to close the gap between QSGD and QSGDinf. QSGD provides theoretical guarantees but is empirically worse than QSGDinf. QSGDinf has no theoretical guarantees. NUQSGD matches the empirical performance of QSGDinf and has slightly stronger asymptotic guarantees than QSGD. We think this progress is worth reporting.
>
> Since submission, we have also improved our understanding of the variance bounds for NUQSGD.
>
> We have proven that, for any given set of levels, there exists a distribution of points with dimension d such that the variance is in Omega(sqrt{d}), and so our bound is tight in d. We will include this proof in the updated version (forthcoming).
>
> Regarding our upper bound and its dependence on s: We have now derived the optimal worst-case variance upper bound for a fixed set of arbitrary levels, expressed as the solution to an integer program with quadratic constraints. We can relax the program to obtain a quadratic program. A coarser analysis yields an upper bound expressed as the solution to a linear program, which is more amenable to analysis.
>
> We are now using these numerical tools to build insight, and will include some plots in the updated draft. For an exponentially spaced collection of levels of the form ((0,p^s, ... , p^2 ,p,1) for p in (0,1) and an integer number of levels, s, we have a numerical method for finding the value p that minimizes the worst-case variance, for any given s and d. We know that our current scheme is near optimal (in worst case) according to the LP bound in some cases. Using these techniques we can get slightly tighter bounds numerically.
>
> >In Figure 3, the experiments only show loss vs. # of iterations, which does not show the actual training time.
>
> Regarding simulation-based learning curves with respect to time, if different compression schemes are run on the same gpu, there will be no difference between any quantization method. This does not hold for error-corrected methods though, since they require additional storage for the error. We will add convergence-versus-time bounds to the updated version.
>
> >In Definition 1, in some cases s is a constant integer, and in some other case become a function, which is very confusing. I also hope the authors can highlight the definition of and , which are essential for understanding the nonuniform quantization mechanism.
>
> In the revision, we clarify these definitions.

---

### Official Review · AnonReviewer3 · 2019-10-23
**Official Blind Review #3**

**Rating:** 6

**Review:**

The authors propose a new scheme for quantizing gradients which are followed by the previous work QSGD [1]. They show that it yields stronger theoretical guarantees than QSGD while showing a great empirical performance.
The main difference between their scheme NUQSGD and QSGD is that they use nonuniform quantization (0, 1/2^{s},  …., 2^{s-1}/2^{s}, 1) instead of uniform quantization (0, 1/s, …, (s-1)/s,1).  Intuitively, by the way, it could reduce quantization error and variance by better matching the properties of normalized vectors.
The results are in 2 parts. First comparing with QSGD, they establish stronger convergence guarantees for NUQSGD, under standard assumptions. They also establish theoretical results for the variance upper bound and expected communication cost of their scheme. Second, they show strong empirical performance on deep models and a large dataset, with an efficient implementation in PyTorch.

However, there are several issues and questions that if fixed or illustrated could be a great paper.

	1) The author claim NUQSGD achieves stronger convergence guarantees comparing with QSGD but hasn't illustrated the point in detail. On page 6, the paragraph named 'NUQSGD vs QSGD' mainly claims that variance upper bound controls the guarantee on the convergence speed by empirically showing the results of variance upper bound. It would be great to include more theoretical analysis which demonstrates the importance of variance upper bound for convergence speed guarantee.
	2) In the experimental part, they control the hyperparameters including batch-size, base learning rate, momentum, and weight decay to be identical with each method. This may cause tuning biases (the setting may favor one method but hurt others' performance).
	3) Although the paper mainly focuses on comparing with QSGD, there are several relative communication efficient training algorithms which I think are worth to compare empirically (at least one of them). For example:
		a. Deep Gradient compression [2]
		b. signSGD [3]
		c. TernGrad [4]
	4) In figure 4, the encoding cost is significantly increased from 4-bit to 8-bit NUQSGD. Any reason why it happens? Is it due to inefficient encoding implementation?

I agree with the authors' point that it's worth to explore the interaction between NUQSGD with more complex reduction patterns like ring-based. Since the ring-based algorithm like all-reduce is more popular in practice nowadays, interacting with it would have a better practical meaning.

[1] D. Alistarh, D. Grubic, J. Z. Li, R. Tomioka, and M. Vojnovic. QSGD: Communication-efﬁcient SGD via gradient quantization and encoding. In Proc. Advances in Neural Information Processing Systems (NIPS), 2017.

[2] Lin Y, Han S, Mao H, Wang Y, Dally WJ. Deep gradient compression: Reducing the communication bandwidth for distributed training. arXiv preprint arXiv:1712.01887. 2017 Dec 5.

[3] Bernstein J, Zhao J, Azizzadenesheli K, Anandkumar A. signSGD with majority vote is communication efficient and fault-tolerant. arXiv. 2018 Oct 11.

[4] W. Wen, C. Xu, F. Yan, C. Wu, Y. Wang, Y. Chen, and H. Li. TernGrad: Ternary gradients to reduce communication in distributed deep learning. In Proc. Advances in Neural Information Processing Systems (NIPS), 2017.

**Experience Assessment:**

I have published one or two papers in this area.

**Review Assessment: Checking Correctness Of Derivations And Theory:**

I assessed the sensibility of the derivations and theory.

**Review Assessment: Checking Correctness Of Experiments:**

I assessed the sensibility of the experiments.

**Review Assessment: Thoroughness In Paper Reading:**

I read the paper at least twice and used my best judgement in assessing the paper.

---

> ### Author Response · Authors · 2019-11-12
> **Response to Review 3**
>
> Thanks for your feedback. Below is our specific feedback to your review. We have also posted a general response (see our top-level comment) to all reviewers addressing high level points.
>
> >It would be great to include more theoretical analysis which demonstrates the importance of variance upper bound for convergence speed guarantee.
>
> We have improved our understanding of the variance bounds for NUQSGD.
>
> We have proven that, for any given set of levels, there exists a distribution of points with dimension d such that the variance is in Omega(sqrt{d}), and so our bound is tight in d. We will include this proof in the updated version (forthcoming).
>
> Regarding our upper bound and its dependence on s: We have now derived the optimal worst-case variance upper bound for a fixed set of arbitrary levels, expressed as the solution to an integer program with quadratic constraints. We can relax the program to obtain a quadratic program. A coarser analysis yields an upper bound expressed as the solution to a linear program, which is more amenable to analysis.
>
> We are now using these numerical tools to build insight, and will include some plots in the updated draft. For an exponentially spaced collection of levels of the form ((0,p^s, ... , p^2 ,p,1) for p in (0,1) and an integer number of levels, s, we have a numerical method for finding the value p that minimizes the worst-case variance, for any given s and d. We know that our current scheme is near optimal (in worst case) according to the LP bound in some cases. Using these techniques we can get slightly tighter bounds numerically.
>
> >In the experimental part, they control the hyperparameters including batch-size, base learning rate, momentum, and weight decay to be identical with each method. This may cause tuning biases (the setting may favor one method but hurt others' performance).
>
> We agree that the performance of each method might slightly improve if we tune hyperparameters for that specific method. However, we are interested in a setting where training is performed with the same standard hyperparameters as those for the full-precision version. We would like to recover full accuracy in this regime. This is the standard set by the original work on QSGD, which is closer to practical applications where hyperparameter tuning is expensive. Again, the goal was to close the empirical performance gap with QSGDinf (we did) and the theoretical gap with QSGD (we did).
>
> >Although the paper mainly focuses on comparing with QSGD, there are several relative communication efficient training algorithms which I think are worth to compare empirically (at least one of them)
>
> Among unbiased schemes, QSGDinf is state-of-the-art but it does not come with theoretical guarantees. QSGD has guarantees but worse performance. Our goal was to close this gap, and we achieved this goal. We think this progress is worth reporting.
>
> We agree that it is interesting to compare NUQSGD with signed-based methods (although we feel this comparison is orthogonal to the problem of closing the performance--theory gap between QSGD and QSGDinf). Recently, error-feedback SGD has been shown to outperform signSGD. We are running experiments comparing NUQSGD with error-corrected methods and hope they are finished before the rebuttal period. Regardless, we will include them in the final paper. One important note is that error-corrected signSGD, sparsified methods, and TernGrad require non-trivial additional parameter tuning to reduce accuracy loss (learning rate. momentum, and warmup tuning---see e.g. "Deep Gradient Compression"). By contrast, our experiments target the setting where training is performed with standard hyperparameters as the full-precision version, and we are able to recover full accuracy in this regime.
>
> >In figure 4, the encoding cost is significantly increased from 4-bit to 8-bit NUQSGD. Any reason why it happens? Is it due to inefficient encoding implementation?
>
> It is because the cost of the compression is proportional to the number of quantization points used, i.e., # quantization points for 8bit = #quantization points for 4bit^2.

---

### Official Review · AnonReviewer2 · 2019-10-24
**Official Blind Review #2**

**Rating:** 3

**Review:**

Brief summary of the paper:
This paper studies data-parallel SGD that K processors work together to minimize an objective function. Each processor computes a stochastic gradient and broadcasts to other peers. In this distributed system, there is a trade-off between the *communication cost* from sharing the stochastic gradient and the *variance* from gradient quantization. This paper is a follow-up of Alistarh et al. (2017). It proposes a non-uniform (logarithmic) quantization scheme (NUQSGD). This paper provides theoretical analysis of the variance and communication cost of NUQSGD. Then the paper analyzes the convergence rate of NUQSGD for convex and smooth objective function. At the end, this paper empirically evaluates NUQSGD for image classification problem.


Originality and significance:
This paper follows up on the parallel SGD framework proposed by Alistarh et al. (2017), where the authors proposed QSGD using a uniform quantization. This paper proposes NUQSGD using a non-uniform quantization method. The quantization of the stochastic gradient amplifies the stochastic variance, which influences the rate of convergence of SGD. Thus, on one hand, it is important to design a quantization method to improve the variance, for the sake of convergence rate. On the other hand, it is also important to decrease the communication cost. NUQSGD does not provide significant improvements in terms of the variance and communication cost.

Theorem 2 and Theorem 3: QSGD has a variance of min {d/s^2, \sqrt{d}/s} and NUQSGD has a variance of min{O(d/2^{-2s}), O(\sqrt{d/2^{-2s}})}. QSGD has communication cost of \tilde O(s(s+\sqrt{d})) and NUQSGD has communication cost of \tilde O(2^{2s}\sqrt{d} ). Compared to QSGD, we can see that NUQSGD improves the dependence on s for the variance term, but it has a worse (exponential) dependence on s for the communication cost. Usually s is a small number and it serves as a hyper-parameter to be tuned. We would expect NUQSGD to improve the dependence on the dimension d, which is more significant. However, NUQSGD has the same dependence on d as QSGD in terms of both variance and communication cost.

Experiments: Figure 3 compares NUQSGD with other parallel SGD algorithms and vanilla SGD. Figure 3 shows how fast the training loss decreases with respect to iterations. It would be great to add learning curves with the ‘time’ being the x-axis as well. Also, I would suggest the authors to record the time needed to proceed one iteration for each parallel algorithm to compare the communication cost.

Quality and clarity:
This paper is well-written.




**Experience Assessment:**

I have read many papers in this area.

**Review Assessment: Checking Correctness Of Derivations And Theory:**

I carefully checked the derivations and theory.

**Review Assessment: Checking Correctness Of Experiments:**

I carefully checked the experiments.

**Review Assessment: Thoroughness In Paper Reading:**

I read the paper thoroughly.

---

> ### Author Response · Authors · 2019-11-12
> **Response to Review 2**
>
> Thanks for your feedback. Below is our specific feedback to your review. We have also posted a general response (see our top-level comment) to all reviewers addressing high level points.
>
> >NUQSGD does not provide significant improvements in terms of the variance and communication cost.
>
> The goal of the paper was to close the gap between QSGD and QSGDinf. QSGD provides theoretical guarantees but is empirically worse than QSGDinf. QSGDinf has no theoretical guarantees. NUQSGD matches the empirical performance of QSGDinf and has slightly stronger asymptotic guarantees than QSGD, and so we don't see the fact that the improvement is "minor" as undermining the significance. In practice, it's much better than QSGD.
>
> >We would expect NUQSGD to improve the dependence on the dimension d, which is more significant
>
> We have proven that, for any given set of levels, there exists a distribution of points with dimension d such that the variance is in Omega(sqrt{d}), and so our bound is tight in d. We will include this proof in the updated version (forthcoming).
>
> >It would be great to add learning curves with the ‘time’ being the x-axis as well. Also, I would suggest the authors to record the time needed to proceed one iteration for each parallel algorithm to compare the communication cost.
>
> Regarding simulation-based learning curves with respect to time, if different compression schemes are run on the same gpu, there will be no difference between any quantization method. This does not hold for error-corrected methods though, since they require additional storage for the error. We will add convergence-versus-time bounds to the updated version. In addition, we will record the time needed to proceed one iteration for each parallel algorithm.

---

### Author Response · Authors · 2019-11-11
**General response to all reviewers**

Dear reviewers,

Thank you for your reviews. In summary, we received the following feedback (key issues):

1. [Variance upper bound; R2,3]. The theoretical improvement over QSGD seems minor. Can stronger theoretical guarantees be obtained? In particular, can you tighten the variance bound in terms of d?

2. [Nonconvexity; R1]. Can convergence results be obtained for nonconvex problems?

3. [Sign-based methods; R1,3]. Is NUQSGD interesting if its performance is comparable to QSGDinf? How does NUQSGD compare with sign-based methods?

4. [Loss vs time; R1,2] How do learning curves look if ‘time’ is the x-axis?

We agree these are important questions. We have a plan to address each of them. We describe that plan below. We hope that if we indeed succeed in executing this plan, you will raise your scores to 8!

We plan to make the following four changes to address the key issues/questions above. If you would require further changes to update your score to 8, please let us know!


*****
**1**
*****
The goal of the paper was to close the gap between QSGD and QSGDinf. QSGD provides theoretical guarantees but is empirically worse than QSGDinf. QSGDinf has no theoretical guarantees. NUQSGD matches the empirical performance of QSGDinf and has slightly stronger asymptotic guarantees than QSGD, and so we don't see the fact that the improvement is "minor" as undermining the significance. In practice, it's much better than QSGD.

That said, we have improved our understanding of the variance bounds for NUQSGD.

Regarding tightness of our variance bounds: Reviewer 1 asks whether we can beat the O(sqrt{d}) dimension dependence in the variance bound. We have proven that, for any given set of levels, there exists a distribution of points with dimension d such that the variance is in Omega(sqrt{d}), and so our bound is tight in d. We will include this proof in the updated version (forthcoming).

Regarding our upper bound and its dependence on s: We have now derived the optimal worst-case variance upper bound for a fixed set of arbitrary levels, expressed as the solution to an integer program with quadratic constraints. We can relax the program to obtain a quadratic program. A coarser analysis yields an upper bound expressed as the solution to a linear program, which is more amenable to analysis.

We are now using these numerical tools to build insight, and will include some plots in the updated draft. For an exponentially spaced collection of levels of the form ((0,p^s, ... , p^2 ,p,1) for p in (0,1) and an integer number of levels, s, we have a numerical method for finding the value p that minimizes the worst-case variance, for any given s and d. We know that our current scheme is near optimal (in worst case) according to the LP bound in some cases. Using these techniques we can get slightly tighter bounds numerically.


*****
**2**
*****
NUQSGD does provide guarantees in the non-convex case as well, since the quantized stochastic gradients are still unbiased. (This is the same for QSGD.) We mention this in the paper, after Theorem 4: "On nonconvex problems, convergence guarantees can be established along the lines of, e.g., (Ghadimi and Lan, 2013, Theorem 2.1)." In particular, this results gives convergence to a second-order stationary point. These are virtually the same guarantees as error-corrected signSGD.  We will include the nonconvex convergence statement in the updated paper.


*****
**3**
*****
Among unbiased schemes, QSGDinf is state-of-the-art but it does not come with theoretical guarantees. QSGD has guarantees but worse performance. Our goal was to close this gap, and we achieved this goal. We think this progress is worth reporting.

We agree that it is interesting to compare NUQSGD with error-corrected methods (although we feel this comparison is orthogonal to the problem of closing the performance--theory gap between QSGD and QSGDinf.) We are running experiments comparing NUQSGD with error-corrected methods and hope they are finished before the rebuttal period. Regardless, we will include them in the final paper. One important note is that error-corrected signSGD, sparsified methods, and TernGrad require non-trivial additional parameter tuning to reduce accuracy loss (learning rate. momentum, and warmup tuning---see e.g. "Deep Gradient Compression"). By contrast, our experiments target the setting where training is performed with standard hyperparameters as the full-precision version, and we are able to recover full accuracy in this regime. This is the standard set by QSGD, which is closer to practical applications.


*****
**4**
*****
Regarding simulation-based learning curves with respect to time, if different compression schemes are run on the same gpu, there will be no difference between any quantization method. This does not hold for error-corrected methods though, since they require additional storage for the error. We will add convergence-versus-time bounds to the updated version.

---

### Author Response · Authors · 2019-11-15
**Summary of changes**

We will be posting a new version of the paper momentarily. This note summarizes the changes:

1. We now report results comparing NUQSGD with error-corrected methods, notably EF-SIGNSGD, on ImageNet. We find that our techniques are superior. In particular, we had to perform significant hyperparameter tuning to even get the error corrected methods (EF-SIGNSGD) to converge. Once we got them to converge, the communication benefits had largely disappeared. We emphasize that our methods achieve full accuracy and speedup under the baseline hyperparameter settings, and do not require additional tuning. This is essential on data sets like ImageNet where tuning is extremely expensive. We also include learning curves when ‘time’ is the x-axis.


2. In the appendix, we prove that, for any given set of levels, there exists a distribution of points with dimension d such that the variance is in Omega(sqrt{d}), and so our bound is tight in d.

3. Regarding our upper bound and its dependence on s: In the appendix, we now derived the optimal worst-case variance upper bounds expressed as an integer QP. We present several relaxations of this bound and plot its dependence on s and d in the appendix.

4.  In the appendix, we now state the implications of our work for convergence on nonconvex problems. As stated in the paper, these results are standard. The important work in this setting is control of the variance and communication cost.

5. We've made various other minor improvements to notation, explanations, etc.

We would welcome suggestions as to what material we might promote to the main body (rather than the appendix). We have left most changes in the appendix to ease the reviewers job in finding these new contributions.

---

### Decision · Program_Chairs · 2019-12-19

**Decision:**

Reject

**Comment:**

This paper proposes a communication-efficient data-parallel SGD with quantization. The method bridges the gap between theory and practice. The QSGD method has theoretical guarantees while QSGDinf doesn't, but the latter gives better result. This paper proves stronger results for QSGD using a different quantization scheme which matches the performance of QSGDinf.

The reviewers find issues with the approach and have pointed some of them out. During the discussion period, we did discuss if reviewers would like to raise their scores. Unfortunately, they still have unresolved issues (see R1's comment).
R1 made another comment recently that they were unable to add to their review:
"The proposed algorithm and the theoretical analysis does not include momentum. However, in the experiments, it is clearly stated that momentum (with a factor of 0.9) is used. Thus, it is unclear whether the experiments really validate the theoretical guarantees. And, it is also unclear how momentum is added for both NUQSGD and EF-SGD, since momentum is not mentioned in Algorithm 1 in this paper, or the paper of QSGD, or the paper of EF-SignSGD. (There is a version of SignSGD with momentum *without* error feedback, called SIGNUM)."

With the current score, the paper does not make the cut for ICLR, but I encourage the authors to revise the paper based on reviewers' feedback. For now, I recommend to reject this paper.